# Clipped SGD Algorithms for Performative Prediction: Tight Bounds for Stochastic Bias and Remedies

Qiang Li [1]   Michal Yemini [2]   Hoi-To Wai [1]

## Abstract

This paper studies the convergence of clipped stochastic gradient descent (SGD) algorithms with decision-dependent data distribution. Our setting is motivated by privacy preserving optimization algorithms that interact with performative data where the prediction models can influence future outcomes. This challenging setting involves the non-smooth clipping operator and non-gradient dynamics due to distribution shifts. We make two contributions in pursuit for a performative stable solution using clipped SGD algorithms. First, we characterize the clipping bias with projected clipped SGD (PCSGD) algorithm which is caused by the clipping operator that prevents PCSGD from reaching a stable solution. When the loss function is strongly convex, we quantify the lower and upper bounds for this clipping bias and demonstrate a bias amplification phenomenon with the sensitivity of data distribution. When the loss function is non-convex, we bound the magnitude of stationarity bias. Second, we propose remedies to mitigate the bias either by utilizing an optimal step size design for PCSGD, or to apply the recent DiceSGD algorithm (Zhang et al., 2024). Our analysis is also extended to show that the latter algorithm is free from clipping bias in the performative setting. Numerical experiments verify our findings.

## 1. Introduction

A recent line of research in statistical learning is to analyze the behavior of stochastic gradient (SGD) type algorithms in tackling stochastic optimization problems with decision-dependent distributions. The latter can be motivated by the training of prediction models under distribution shifts (Quiñonero-Candela et al., 2022) where the data may 'react' to the changing prediction models. A common application scenario is that the training data involve human input that responds strategically to the model (Hardt et al., 2016). Distribution shifts affect the convergence of SGD-type algorithms and their efficacy as the distributions of gradient estimates vary gradually. The modeling of such behavior has led to the *performative prediction problem* (Perdomo et al., 2020); see the recent overview (Hardt & Mendler-Dünner, 2023).

For the performative prediction problem, there is a growing literature on developing stochastic algorithms to find a fixed point solution of the repeated risk minimization procedure, also known as the performative stable (PS) solution(s). In the absence of knowledge of the distribution shift, the latter is the natural solution obtained by stochastic approximation algorithms. In particular, most prior works have only focused on settings when the loss function is strongly convex, e.g., (Mendler-Dünner et al., 2020; Drusvyatskiy & Xiao, 2023; Brown et al., 2022; Li & Wai, 2022). When the loss function is non-convex, the existing results are much rarer. To list a few, (Mofakhami et al., 2023) assumes a least-square like loss function; (Li & Wai, 2024) provided guarantees for general non-convex performative prediction. As an alternative, prior works also studied stochastic algorithms for finding performative optimal solutions by an extra step that estimates the form of distribution shifts (Izzo et al., 2021; Miller et al., 2021; Narang et al., 2023).

However, most of the existing works in the performative prediction literature have focused on analyzing SGD algorithms admitting a smooth drift term. An open problem in the performative prediction literature is to analyze the behavior of *clipped SGD* algorithms which limit the magnitude of the stochastic gradient at every update step; this in turn distorts the considered distribution shift compared with the nominal dynamics with no clipping. Gradient clipping is used to deal with multiple obstacles in learning algorithms such as the need for privacy preservation (Abadi et al., 2016), dealing with gradient explosion in non-smooth learning (Shor, 2012) such as training neural networks (Mikolov et al., 2012; Zhang et al., 2020a), solving quasi-convex problems (Hazan et al., 2015), etc. Despite the difficulty with treating the non-

[1]Department of System Engineering and Engineering Management, The Chinese University of Hong Kong, Hong Kong [2]Faculty of Engineering, Bar-Ilan University, Israel. Correspondence to: Hoi-To Wai <htwai@se.cuhk.edu.hk>.

*Proceedings of the 42nd International Conference on Machine Learning*, Vancouver, Canada. PMLR 267, 2025. Copyright 2025 by the author(s).

smooth drift term with clipping, the clipped SGD algorithm has been analyzed in a multitude of works in the standard i.i.d. sampling setting with non-decision-dependent data. In fact, (Mai & Johansson, 2021) studied the non-smooth optimization setting, and (Gorbunov et al., 2020) analyzed the convergence under heavy-tailed noise. On the other hand, it was found in (Chen et al., 2020) that the clipped SGD may exhibit an asymptotic bias, i.e., deviation from the optimal/stationary solution, for asymmetric gradient distribution, and later on (Koloskova et al., 2023) showed that the bias can be unavoidable. This led to several recent works which considered *debiasing* clipped SGD, e.g., (Khirirat et al., 2023) studied bias-free clipped SGD for distributed optimization, (Zhang et al., 2024) studied a differentially private SGD algorithm using two clipping operators simultaneously.

The current paper aims to study the convergence of clipped SGD algorithms in the *performative prediction* setting. We focus on the interplay between the distribution shift and the non-smooth clipped updates. Our main contributions are:

- We show that **PCSGD** converges in expectation to a neighborhood of the performative stable solution, a fixed point studied by Perdomo et al. (2020). For strongly convex losses, while the convergence rate is $\mathcal{O}(1/t)$ where $t$ is the iteration number, we show that the clipping operation induces an *asymptotic clipping bias*. For non-convex losses, we show that in $T$ iterations, the scheme converges at the rate of $\mathcal{O}(1/\sqrt{T})$ towards a biased stationary performative stable solution. In both cases, we show that the magnitude of bias is proportional to the sensitivity of distribution shift and clipping threshold.

- For the case of strongly convex losses, we further show that there exists a matching lower bound for the asymptotic clipping bias upon specifying the class of performative risk optimization problem. Together with the derived upper bound, we demonstrate a *bias amplification* effect of **PCSGD** when subject to distribution shift in performative prediction.

- As a remedy to the bias effect of **PCSGD**, we study the recently proposed **DiceSGD** algorithm (Zhang et al., 2024). We show that with a doubly clipping mechanism on both the gradient and clipping error, the algorithm can converge exactly (nearly) to the PS solution under strongly convex (non-convex) loss.

Further, we show that there exists a tradeoff between the differential privacy guarantee and computation complexity that affects the optimal step size selection. The paper is organized as follows. §2 introduces the performative prediction problem and the **PCSGD** algorithm. §3 presents the theoretical analysis for **PCSGD**. §4 discusses the **DiceSGD** algorithm and how it mitigates the clipping bias. Our analysis includes both strongly convex and non-convex settings.

§5 presents the numerical experiments.

**Notations.** Let $\mathbb{R}^d$ be the $d$-dimensional Euclidean space equipped with inner product $\langle \cdot \,|\, \cdot \rangle$ and induced norm $\|x\| = \sqrt{\langle x \,|\, x \rangle}$. $\mathbb{E}[\cdot]$ denotes taking expectation w.r.t all randomness, $\mathbb{E}_t[\cdot] := \mathbb{E}_t[\cdot|\mathcal{F}_t]$ means taking conditional expectation on filtration $\mathcal{F}_t := \sigma(\{\boldsymbol{\theta}_0, \boldsymbol{\theta}_1, \cdots, \boldsymbol{\theta}_t\})$, where $\sigma(\cdot)$ is the sigma-algebra generated by the random variables in the operand.

## 2. Problem Setup

This section introduces the performative prediction problem and a simple projected clipped SGD algorithm. Our goal is to learn a prediction/classification model $\boldsymbol{\theta} \in \mathcal{X}$ via the stochastic optimization problem:

$$\min_{\boldsymbol{\theta} \in \mathcal{X}} \ \mathbb{E}_{Z \sim \mathcal{D}(\boldsymbol{\theta})}[\ell(\boldsymbol{\theta}; Z)], \tag{1}$$

where $\mathcal{X} \subseteq \mathbb{R}^d$ is a closed convex set and $\ell(\cdot)$ is differentiable w.r.t. $\boldsymbol{\theta}$.

An intriguing feature of (1) is that the optimization problem is defined along with a *decision-dependent distribution* $\mathcal{D}(\boldsymbol{\theta})$ where the distribution of the sample $Z \sim \mathcal{D}(\boldsymbol{\theta})$ depends on $\boldsymbol{\theta}$. For example, it may take the form of the best response for a utility function parameterized by $\boldsymbol{\theta}$. This setup models a scenario where the prediction model may influence the outcomes it aims to predict, also known as the *performative prediction* problem; see (Perdomo et al., 2020; Hardt & Mendler-Dünner, 2023).

The challenge in tackling (1) lies in that the decision variable $\boldsymbol{\theta}$ appears in both the loss function $\ell(\boldsymbol{\theta}; Z)$ and the distribution $\mathcal{D}(\boldsymbol{\theta})$. As a result, (1) is in general *non-convex* even if $\ell(\cdot)$ is (strongly) convex. To this end, a remedy is to study the fixed point solutions deduced from tackling the partial optimization of minimizing $\mathbb{E}_{Z \sim \mathcal{D}(\overline{\boldsymbol{\theta}})}[\ell(\boldsymbol{\theta}; Z)]$ w.r.t. $\boldsymbol{\theta}$ when the distribution depends on a fixed $\overline{\boldsymbol{\theta}}$.

When $\ell(\cdot)$ is strongly convex in $\boldsymbol{\theta}$, a popular solution concept is the performative stable (PS) solution (Perdomo et al., 2020):

**Definition 1.** *The solution* $\boldsymbol{\theta}_{PS} \in \mathcal{X}$ *is called a PS solution to* (1) *if it satisfies*

$$\boldsymbol{\theta}_{PS} = \arg\min_{\boldsymbol{\theta} \in \mathcal{X}} \ \mathbb{E}_{Z \sim \mathcal{D}(\boldsymbol{\theta}_{PS})}[\ell(\boldsymbol{\theta}; Z)]. \tag{2}$$

Note that $\boldsymbol{\theta}_{PS}$ is unique and well-defined provided that (i) the loss function is smooth, (ii) $\mathcal{D}(\boldsymbol{\theta})$ is not overly sensitive to shifts in $\boldsymbol{\theta}$; see §3.1 for details.

Alternatively, when $\ell(\cdot)$ is non-convex in $\boldsymbol{\theta}$ and $\mathcal{X} \equiv \mathbb{R}^d$, a recent solution concept is the stationary PS (SPS) solution (Li & Wai, 2024):

**Definition 2.** *The solution* $\boldsymbol{\theta}_{SPS} \in \mathcal{X}$ *is called a stationary PS solution to* (1) *if it satisfies*

$$\mathbb{E}_{Z \sim \mathcal{D}(\boldsymbol{\theta}_{SPS})}[\nabla \ell(\boldsymbol{\theta}_{SPS}; Z)] = \mathbf{0}. \tag{3}$$

Note that if $\ell(\cdot)$ is strongly convex, (3) will recover the definition of PS solution in (2).

It is clear that $\boldsymbol{\theta}_{PS}$, $\boldsymbol{\theta}_{SPS}$ do not solve (1), nor are they stationary solutions of (1). However, they remain reasonable estimates to the solutions of (1). As shown in (Perdomo et al., 2020, Theorem 4.3), the disparity between $\boldsymbol{\theta}_{PS}$ and the optimal solution to (1) is bounded by the sensitivity of the decision-dependent distribution (cf. A5).

To search for $\boldsymbol{\theta}_{PS}$ or $\boldsymbol{\theta}_{SPS}$, SGD-based schemes with greedy deployment, i.e., the learner deploys the latest model at the population after every SGD update, have been widely deployed, see (Mendler-Dünner et al., 2020; Drusvyatskiy & Xiao, 2023; Li & Wai, 2024).

Motivated by privacy preserving optimization (Abadi et al., 2016) and improved stability in training deep neural networks (Zhang et al., 2020a), in this paper we are interested in SGD algorithms which clip the stochastic gradient at each update. With the greedy deployment scheme in mind, our first objective is to study the projected clipped SGD (**PCSGD**) algorithm for performative prediction:

**Deploy:** $Z_{t+1} \sim \mathcal{D}(\boldsymbol{\theta}_t),$ (4)

**Update:** $\boldsymbol{\theta}_{t+1} = \mathcal{P}_{\mathcal{X}}\big(\boldsymbol{\theta}_t - \gamma_{t+1}\mathsf{clip}_c(\nabla\ell(\boldsymbol{\theta}_t; Z_{t+1}))\big),$ (5)

where $\mathcal{P}_{\mathcal{X}}(\cdot)$ denotes the Euclidean projection operator onto $\mathcal{X}$, and $\mathsf{clip}_c(\cdot)$ is the clipping operator: for any $\boldsymbol{g} \in \mathbb{R}^d$,

$$\mathsf{clip}_c(\boldsymbol{g}): \boldsymbol{g} \in \mathbb{R}^d \mapsto \min\{1, c/\|\boldsymbol{g}\|_2\}\,\boldsymbol{g},$$ (6)

such that $c > 0$ is a clipping parameter. Notice that if $c \to \infty$, (4) is reduced to the projected SGD algorithm for performative prediction analyzed in (Mendler-Dünner et al., 2020; Drusvyatskiy & Xiao, 2023). Additionally, when $\mathcal{D}(\boldsymbol{\theta})$ is independent of $\boldsymbol{\theta}$, i.e., $\mathcal{D}(\boldsymbol{\theta}) = \mathcal{D}$ for all $\boldsymbol{\theta} \in \mathcal{X}$, the **PCSGD** algorithm does not exhibit a distribution shift and coincides with a projected version of the standard clipped SGD algorithm examined in (Koloskova et al., 2023). Thus, our model and results extend both (Mendler-Dünner et al., 2020; Drusvyatskiy & Xiao, 2023) and (Koloskova et al., 2023).

For $c < \infty$ and especially when $c < \|\nabla\ell(\boldsymbol{\theta}_t; Z_{t+1})\|$, the **PCSGD** recursion pertains to a non-gradient dynamics with non-smooth drifts due to the clipping operator; see Sec. 3. Prior analysis of **PCSGD** are no longer applicable in this scenario.

**Remark 1.** *Under the performative prediction setting, (Drusvyatskiy & Xiao, 2023) considered an alternative clipping model which approximates the loss function $\ell(\boldsymbol{\theta}; z)$ by a linear model $\ell_{\boldsymbol{\theta}'}(\boldsymbol{\theta}; z) := \ell(\boldsymbol{\theta}'; z) + \langle\nabla\ell(\boldsymbol{\theta}'; z) \,|\, \boldsymbol{\theta} - \boldsymbol{\theta}'\rangle$, then updates $\boldsymbol{\theta}$ by applying a proximal gradient method on $\ell_{\boldsymbol{\theta}}(\boldsymbol{\theta}; z)$ within a bounded set. This clips the models under training $\boldsymbol{\theta}$ instead of clipping $\nabla\ell(\boldsymbol{\theta}; z)$ as in (4). Such algorithm belongs to the class of model based gradient methods whose fixed point is $\boldsymbol{\theta}_{PS}$ when $\ell(\cdot)$ is strongly convex.*

## 3. Main Results for PCSGD

In pursuit for a stochastic algorithm that finds $\boldsymbol{\theta}_{PS}$ or $\boldsymbol{\theta}_{SPS}$, we first study the convergence properties of **PCSGD**. Additionally, we examine the tradeoff between model efficacy and privacy preservation of the algorithm.

### 3.1. Analysis of the PCSGD Algorithm

The analysis of (4) involves challenges that are unique to the decision-dependent distributions. Curiously, the expectation of the *unclipped* stochastic gradient $\nabla\ell(\boldsymbol{\theta}_t; Z_{t+1})$ is not a gradient. To see this, consider the special case of

$$\ell(\boldsymbol{\theta}; Z) = (1/2)\|\boldsymbol{\theta} - Z\|^2$$

and $Z \sim \mathcal{D}(\boldsymbol{\theta}) \Leftrightarrow Z \sim \mathcal{N}(\boldsymbol{A}\boldsymbol{\theta}; \boldsymbol{I})$. Observe that $\mathbb{E}_t[\nabla\ell(\boldsymbol{\theta}_t; Z_{t+1})] = (\boldsymbol{I} - \boldsymbol{A})\boldsymbol{\theta}_t$ has a Jacobian of $\boldsymbol{I} - \boldsymbol{A}$ which is asymmetric whenever $\boldsymbol{A}$ is asymmetric. Analyzing **PCSGD** requires studying a non-gradient dynamics with non-smooth drifts induced by the clipping operator. In the subsequent discussion, we analyze **PCSGD** through identifying a suitable Lyapunov function depending on properties of the loss function $\ell(\cdot)$.

We define the shorthand notation:

$$f(\boldsymbol{\theta}_1, \boldsymbol{\theta}_2) := \mathbb{E}_{Z\sim\mathcal{D}(\boldsymbol{\theta}_2)}[\ell(\boldsymbol{\theta}_1; Z)].$$ (7)

Unless otherwise specified, the vector $\nabla f(\boldsymbol{\theta}_1; \boldsymbol{\theta}_2)$ refers to the gradient taken w.r.t. the first argument $\boldsymbol{\theta}_1$. We begin by stating a few assumptions pertaining to the performative prediction problem (1):

**A1.** *For any $\bar{\boldsymbol{\theta}} \in \mathcal{X}$, the function $f(\boldsymbol{\theta}; \bar{\boldsymbol{\theta}})$ is $\mu$ strongly convex w.r.t. $\boldsymbol{\theta}$, i.e., for any $\boldsymbol{\theta}', \boldsymbol{\theta} \in \mathcal{X}$,*

$$f(\boldsymbol{\theta}'; \bar{\boldsymbol{\theta}}) \geq f(\boldsymbol{\theta}; \bar{\boldsymbol{\theta}}) + \langle\nabla f(\boldsymbol{\theta}; \bar{\boldsymbol{\theta}}) \,|\, \boldsymbol{\theta}' - \boldsymbol{\theta}\rangle + \frac{\mu}{2}\|\boldsymbol{\theta}' - \boldsymbol{\theta}\|^2.$$

**A2.** *The gradient map $\nabla\ell(\cdot; \cdot)$ is $L$-Lipschitz, i.e., for any $\boldsymbol{\theta}_1, \boldsymbol{\theta}_2 \in \mathcal{X}$, $z_1, z_2 \in \mathsf{Z}$,*

$$\|\nabla\ell(\boldsymbol{\theta}_1; z_1) - \nabla\ell(\boldsymbol{\theta}_2; z_2)\| \leq L\big(\|\boldsymbol{\theta}_1 - \boldsymbol{\theta}_2\| + \|z_1 - z_2\|\big)$$

*Moreover, there exists a constant $\ell^\star > -\infty$ such that $\ell(\boldsymbol{\theta}; z) \geq \ell^\star$ for any $\boldsymbol{\theta} \in \mathcal{X}$.*

The above assumptions are common in the literature, e.g., (Perdomo et al., 2020; Mendler-Dünner et al., 2020; Drusvyatskiy & Xiao, 2023). In addition, we require that

**A 3.** *There exists a constant $G \geq 0$ such that $\sup_{\boldsymbol{\theta}\in\mathcal{X}, z\in\mathsf{Z}} \|\nabla\ell(\boldsymbol{\theta}; z)\| \leq G$.*

This condition can be satisfied if $\mathcal{X}$ is compact; or for cases such as the sigmoid loss functions. Notice that a similar condition is used in (Zhang et al., 2020a). In some cases, we will use the following standard variance condition to obtain a tighter bound:

**A4.** *There exists constants $\sigma_0, \sigma_1 \geq 0$ such that for any $\boldsymbol{\theta}_1, \boldsymbol{\theta}_2 \in \mathcal{X}$, it holds*

$$\mathbb{E}_{Z \sim \mathcal{D}(\boldsymbol{\theta}_2)} \left[ \|\nabla \ell(\boldsymbol{\theta}_1; Z) - \nabla f(\boldsymbol{\theta}_1; \boldsymbol{\theta}_2)\|^2 \right]$$
$$\leq \sigma_0^2 + \sigma_1^2 \|\nabla f(\boldsymbol{\theta}_1; \boldsymbol{\theta}_2)\|^2.$$

**Strongly Convex Loss** We first discuss the convergence of **PCSGD** towards $\boldsymbol{\theta}_{PS}$ under A1. We provide both *upper* and *lower* bounds for the asymptotic bias of **PCSGD**, and demonstrate a *bias amplification* effect as the sensitivity parameter of the distribution $\beta$ increases (cf. A5). Our result gives the first tight characterization of the bias phenomena in the literature.

To establish the convergence of **PCSGD** in this case, we will need the following additional assumptions:

**A5.** *There exists $\beta \geq 0$ such that*

$$W_1(\mathcal{D}(\boldsymbol{\theta}), \mathcal{D}(\boldsymbol{\theta}')) \leq \beta \|\boldsymbol{\theta} - \boldsymbol{\theta}'\|, \ \forall \, \boldsymbol{\theta}, \boldsymbol{\theta}' \in \mathcal{X}.$$

*Notice that*

$$W_1(\cdot, \cdot) = \inf_{J \in \mathcal{J}(\cdot, \cdot)} \mathbb{E}_{(z, z') \sim J} \left[ \|z - z'\|_1 \right]$$

*is the Wasserstein-1 distance, where $\mathcal{J}(\mathcal{D}(\boldsymbol{\theta}), \mathcal{D}(\boldsymbol{\theta}'))$ is the set of all joint distributions on $\mathsf{Z} \times \mathsf{Z}$ whose marginal distributions are $\mathcal{D}(\boldsymbol{\theta}), \mathcal{D}(\boldsymbol{\theta}')$.*

We emphasize that $\beta$ in A5 quantifies the *sensitivity* of the distribution against perturbation with respect to the decision model $\boldsymbol{\theta}$. It will play an important role for the analysis below. Notice that A1, 2, 5 imply that $\|\boldsymbol{\theta}^\star - \boldsymbol{\theta}_{PS}\| \leq \frac{2L\beta}{\mu}$ where $\boldsymbol{\theta}^\star \in \mathrm{Arg}\min_{\boldsymbol{\theta} \in \mathcal{X}} \mathbb{E}_{Z \sim \mathcal{D}(\boldsymbol{\theta})}[\ell(\boldsymbol{\theta}; Z)]$ is an optimal solution to the performative risk minimization problem (Perdomo et al., 2020).

The first result upper bounds the squared norm of the error $\hat{\boldsymbol{\theta}}_t := \boldsymbol{\theta}_t - \boldsymbol{\theta}_{PS}$ at the $t$th iteration of **PCSGD** in expectation:

**Theorem 3.** *(Upper bound) Under A1, 2, 3, 5. Suppose that $\beta < \frac{\mu}{L}$, the step sizes $\{\gamma_t\}_{t \geq 1}$ are non-increasing and satisfy i) $\frac{\gamma_{t-1}}{\gamma_t} \leq 1 + \frac{\mu - L\beta}{2}\gamma_t$, and ii) $\gamma_t \leq \frac{2}{\mu - L\beta}$. Then, for any $t \geq 1$, the expected squared distance between $\boldsymbol{\theta}_t$ and the performative stable solution $\boldsymbol{\theta}_{PS}$ satisfies*

$$\mathbb{E}\|\hat{\boldsymbol{\theta}}_{t+1}\|^2 \leq \prod_{i=1}^{t+1}(1 - \tilde{\mu}\gamma_i)\|\hat{\boldsymbol{\theta}}_0\|^2 + \frac{2c_1}{\tilde{\mu}}\gamma_{t+1} + \frac{8\mathcal{C}_1}{\tilde{\mu}^2}, \quad (8)$$

*where $c_1 := 2(c^2 + G^2)$, $\mathcal{C}_1 := (\max\{G - c, 0\})^2$, and $\tilde{\mu} := \mu - L\beta$.*

The proof is relegated to §B. We remark that A3 assumes that the stochastic gradient is uniformly bounded, where $G$ also accounts for the variance of stochastic gradient. If the additional assumption such as A4 holds, it can be proven

that (8) holds with $c_1 = \mathcal{O}(c^2 + \sigma_0^2)$. Our bound highlights the dependence on $t$ and the distribution shift parameter $\beta$.

From (8), as $t \to \infty$, with a properly tuned step size, the first term decays sub-exponentially to zero, the second term also converges to zero if $\gamma_t = \mathcal{O}(1/t)$. Regardless of the step size choice, the last term never vanish. It indicates an asymptotic clipping bias of **PCSGD** and coincides with the observation in (Koloskova et al., 2023) for non-decision-dependent distribution.

As $t \to \infty$, the bound in (8) converges to a non-vanishing clipping bias term that reads

$$\mathtt{Bias} = 8\mathcal{C}_1/\tilde{\mu}^2 = \mathcal{O}(1/(\mu - L\beta)^2). \quad (9)$$

Even when $\beta = 0$, this bias term is improved over (Zhang et al., 2020a) which obtained an $\mathcal{O}(1/c)$ scaling. It turns out that the above characterization of the bias is *tight* w.r.t. $\mu - L\beta$. We observe:

**Theorem 4.** *(Lower bound) For any clipping threshold $c \in (0, G)$, there exists a function $\ell(\boldsymbol{\theta}; Z)$ and a decision-dependent distribution $\mathcal{D}(\boldsymbol{\theta})$ satisfying A1, 2, 3, 5, such that for all fixed-points of **PCSGD** $\boldsymbol{\theta}_\infty$ satisfying $\mathbb{E}_{Z \sim \mathcal{D}(\boldsymbol{\theta}_\infty)}[\mathsf{clip}_c(\nabla \ell(\boldsymbol{\theta}_\infty; Z))] = \mathbf{0}$, it holds that*

$$\|\boldsymbol{\theta}_\infty - \boldsymbol{\theta}_{PS}\|^2 = \Omega\left(1/(\mu - L\beta)^2\right). \quad (10)$$

The proof is relegated to §C.

Provided that $\beta < \frac{\mu}{L}$, Theorems 3 and 4 show that **PCSGD** admits a clipping bias[1] of $\Theta(1/(\mu - L\beta)^2)$. It illustrates a *bias amplification* effect where as the *sensitivity level* of distribution $\beta$ increases, the bias will increase as $\beta \uparrow \frac{\mu}{L}$. This is a unique phenomenon to performative prediction where in addition to the clipping level, the data distribution contributes to the bias.

**Non-convex Loss** Next, we discuss the convergence of **PCSGD** when the loss function is smooth but possibly non-convex, i.e., without A1.

Our study concentrates on the case where $\mathcal{X} \equiv \mathbb{R}^d$ such that the projection operator is equivalent to an identity operator. We consider the following condition:

**A6.** *There exists $\beta \geq 0$ such that*

$$d_{TV}(\mathcal{D}(\boldsymbol{\theta}), \mathcal{D}(\boldsymbol{\theta}')) \leq \beta \|\boldsymbol{\theta} - \boldsymbol{\theta}'\|, \ \forall \, \boldsymbol{\theta}, \boldsymbol{\theta}' \in \mathcal{X}.$$

*where $d_{TV}(\mathcal{D}(\boldsymbol{\theta}), \mathcal{D}(\boldsymbol{\theta}'))$ denotes the total variation (TV) distance between the distributions $\mathcal{D}(\boldsymbol{\theta}), \mathcal{D}(\boldsymbol{\theta}')$.*

The interpretation of $\beta$ is similar to that of A5. Notice that as $d_{TV}(\mu, \upsilon) \geq W_1(\mu, \upsilon)$, A6 yields a stronger requirement on the sensitivity of the distribution shift than A5 in general. Moreover, we require that

---

[1]The lower bound in Theorem 4 holds for any $\beta \geq 0$. When $\beta \geq \frac{\mu}{L}$, **PCSGD** may not converge.

**A 7.** *There exists a constant $\ell_{max} \geq 0$ such that $\sup_{\boldsymbol{\theta} \in \mathbb{R}^d, z \in \mathsf{Z}} |\ell(\boldsymbol{\theta}; z)| \leq \ell_{max}$.*

The above condition can be satisfied in practical scenarios where (1) involves the training of nonlinear models such as neural networks with bounded outputs.

Under the above conditions, we observe the following upper bound on the SPS measure in Definition 2 for the clipped SGD algorithm:

**Theorem 5.** *Under A2, 3, 4, 6, 7. Let the step sizes satisfy $\sup_{t \geq 1} \gamma_t \leq \frac{1}{2(1+\sigma_1^2)}$. Then, for any $T \geq 1$, the iterates $\{\boldsymbol{\theta}_t\}_{t \geq 0}$ generates by (4) satisfy:*

$$\sum_{t=0}^{T-1} \gamma_{t+1} \mathbb{E}\left[\|\nabla f(\boldsymbol{\theta}_t; \boldsymbol{\theta}_t)\|^2\right] \leq 8\Delta_0 + 4L\sigma_0^2 \sum_{t=0}^{T-1} \gamma_{t+1}^2$$
$$+ 8\mathsf{b}(\beta, c) \sum_{t=0}^{T-1} \gamma_{t+1}, \quad (11)$$

*where $\Delta_0 := \mathbb{E}[f(\boldsymbol{\theta}_0; \boldsymbol{\theta}_0) - \ell^\star]$ is an upper bound to the initial optimality gap for performative risk, and*

$$\mathsf{b}(\beta, c) = \ell_{max}\beta(\sigma_0 + 8(1+\sigma_1^2)\ell_{max}\beta) + 2\max\{G-c, 0\}^2.$$

The proof is relegated to §D.

To get further insights, fix any $T \geq 1$ and set a constant step size $\gamma_t = 1/\sqrt{T}$, we let $\mathsf{T}$ be a random variable chosen uniformly and independently from $\{0, 1, \cdots, T-1\}$. The iterates satisfy:

$$\mathbb{E}\left[\|\nabla f(\boldsymbol{\theta}_\mathsf{T}; \boldsymbol{\theta}_\mathsf{T})\|^2\right] \leq \left(\Delta_0 + \frac{L\sigma_0^2}{2}\right)\frac{8}{\sqrt{T}} + 8\,\mathsf{b}(\beta, c).$$

We observe that the first term vanishes as $T \to \infty$. The second term represents an upper bound to the asymptotic bias for the clipped SGD algorithm that scales with the distribution shift's sensitivity $\beta$ and the clipping threshold $c$. Notice that as shown in (Koloskova et al., 2023), the clipped SGD algorithm admits a non-zero asymptotic bias for the case of non-convex optimization. In comparison, our bound can be directly controlled by the clipping threshold.

Compared to the findings for the strongly convex case, while the asymptotic bias persists in Theorem 5 and it also depends on $\beta$ and $\max\{G - c, 0\}$, the latter effects are combined in an *additive* fashion. Nevertheless, we suspect that this bound on the bias can be improved in the non-convex case. We note that in general, finding a tight lower bound for the convergence of clipped SGD in the non-convex, decision-dependent setting is an open problem.

### 3.2. Differential Privacy Guarantees

Our next objective is to study the implications of the convergence analysis on the privacy preservation power of **PCSGD**.

To fix idea, we first introduce the definition of $(\varepsilon, \delta)$ *differential privacy (DP)* measure which is customary for measuring the level of privacy leakage of a stochastic algorithm:

**Definition 6.** *(Dwork & Roth, 2014) A randomized mechanism $\mathcal{M} : \mathcal{D} \mapsto \mathcal{R}$ satisfies $(\varepsilon, \delta)-$differential privacy if for any two adjacent inputs $\mathsf{D}, \mathsf{D}' \in \mathcal{D}$ which differs by only 1 different sample, and for any subset of outputs $S \subseteq \mathcal{R}$,*

$$\Pr[\mathcal{M}(\mathsf{D}) \in S] \leq e^\varepsilon \Pr[\mathcal{M}(\mathsf{D}') \in S] + \delta. \quad (12)$$

The definition can be understood through the case with $\varepsilon, \delta \approx 0$. In such case, the output (which consists of the entire training history) of an DP algorithm on two adjacent databases, $\mathsf{D}_0, \mathsf{D}_0'$ will be indistinguishable, thus protecting the identity of each data sample.

To discuss privacy preservation in the framework of (Abadi et al., 2016) using Definition 6, we need to introduce a few specifications on the performative prediction problem (1) and modifications to the **PCSGD** algorithm. First, we consider a *fixed, finite database* setting for (1). The database consists of $m$ samples $\mathsf{D}_0 := \{\bar{z}_i\}_{i=1}^m$. The decision-dependent distribution $\mathcal{D}(\boldsymbol{\theta})$ is defined accordingly: $Z \sim \mathcal{D}(\boldsymbol{\theta})$ refers to a (batch of) sample drawn from the database $\mathsf{D}_0$ with *distribution shift* governed by a random map $\mathcal{S}_i : \mathcal{X} \to \mathsf{Z}$. Let $s_i(\boldsymbol{\theta})$ be a realization of $\mathcal{S}_i(\boldsymbol{\theta})$,

$$Z = \bar{z}_i + s_i(\boldsymbol{\theta}), \quad i \sim \text{Unif}([m]). \quad (13)$$

Second, we concentrate on a slightly modified version of (4) proposed in (Abadi et al., 2016): set $Z_{t+1} \sim \mathcal{D}(\boldsymbol{\theta}_t)$,

$$\boldsymbol{\theta}_{t+1} = \mathcal{P}_\mathcal{X}\big(\boldsymbol{\theta}_t - \gamma_{t+1}(\text{clip}_c(\nabla\ell(\boldsymbol{\theta}_t; Z_{t+1})) + \zeta_{t+1})\big), \quad (14)$$

for $t = 0, ..., T-1$, where $\zeta_{t+1} \sim \mathcal{N}(\mathbf{0}, \sigma_{\mathsf{DP}}^2\boldsymbol{I})$ is an artificial (Gaussian) noise added to preserve privacy. Compared to approaches such as (Chaudhuri et al., 2011) which directly add Laplacian noise to SGD, the algorithm with clipping offers better numerical stability.

Overall, we observe that (14) is *a randomized mechanism* applied on the *fixed dataset* $\mathsf{D}_0$, where the distribution shifts is treated as a part of the mechanism. In fact, we can analyze the DP measure of **PCSGD** using the following corollary of (Abadi et al., 2016):

**Corollary 1.** *(Privacy Guarantee) For any $\varepsilon \leq T/m^2$, $\delta \in (0, 1)$, and $c > 0$, the **PCSGD** algorithm with greedy deployment is $(\varepsilon, \delta)$-DP after $T$ iterations if we let $\sigma_{\mathsf{DP}} = c\sqrt{T\log(1/\delta)}/(m\varepsilon)$.*

See §E for detailed proof. Corollary 1 states that **PCSGD** can achieve $(\varepsilon, \delta)$-DP with an appropriate DP noise level. Furthermore, the $\sqrt{T}$ dependence for $\sigma_{\mathsf{DP}}$ leads to an additional source of bias in (8). Together with Theorem 3, the corollary gives a guideline for setting the algorithm's parameters such as clipping threshold $c$ and step size $\gamma$.

We are now equipped with the machinery to study the implications of previous convergence results with respect to the DP guarantees for strongly convex $\ell(\cdot)$. Suppose that the DP parameters $(\varepsilon, \delta)$ are fixed. From Corollary 1, the DP noise variance $\sigma_{\mathsf{DP}}^2$ is proportional to $T$. It follows that increasing $T$ leads to an increase in the error term $c_1$ observed in Theorem 3 and it adds to the bias even when $\gamma_t = \mathcal{O}(1/T)$. In this light, the next result demonstrates how to design an optimal constant step size that minimizes the upper bound in (8) corresponding to the efficacy of the model:

**Corollary 2.** *(Finite-time Analysis) Fix the privacy parameters at $(\varepsilon, \delta)$ and clipping threshold at $c$. Assume that $G > c$ and a constant step size is used in* **PCSGD**. *For all $T \geq 1$, to achieve the minimum for $\mathbb{E}\|\hat{\boldsymbol{\theta}}_T\|^2$, the optimal constant step size can be set as*

$$\gamma^\star = \frac{\log \Delta(\tilde{\mu})^{-1}}{\tilde{\mu} T}, \text{ where } \Delta(\tilde{\mu}) := \frac{2(2(c^2+G^2)+d\sigma_{\mathsf{DP}}^2)}{T \tilde{\mu}^2 \|\hat{\boldsymbol{\theta}}_0\|^2}. \quad (15)$$

*Let $\phi := \frac{d \log(1/\delta)}{m^2 \varepsilon^2}$, then (8) simplifies to*

$$\mathbb{E}\left\|\hat{\boldsymbol{\theta}}_T\right\|^2 = \mathcal{O}\left(\frac{\mathcal{C}_1}{\tilde{\mu}^2} + \left[\frac{c^2+G^2}{T\tilde{\mu}^2} + \frac{c^2 \phi}{\tilde{\mu}^2}\right] \log\left(\frac{\tilde{\mu}^2}{\phi c^2}\right)\right).$$

Besides the $\mathcal{O}(1/T)$ dependence, we observe that $\gamma^\star$ is affected by the sensitivity parameter through $\tilde{\mu}^2$, and the DP parameters $\varepsilon, \delta$.

Lastly, we examine the case when $T \gg 1$. Observe that by setting $\gamma_t = c_2/\tilde{\mu} T$ for any $c_2 > 1$, the first term in (8) vanishes with sufficiently large $T$. Similar to the above corollary, enforcing DP guarantee leads to an asymptotic bias in (8) which depends on the interplay between the clipping threshold $c$, DP noise variance $\sigma_{\mathsf{DP}}^2$, etc. We obtain the following asymptotic guarantee upon optimizing the clipping threshold $c$:

**Corollary 3.** *(Asymptotic Analysis) Fix the privacy parameters at $(\varepsilon, \delta)$. Let $T \gg 1$ and set $\gamma_t = (1 + c_2)/(\tilde{\mu} T)$, the optimum asymptotic upper bound for the deviation from $\boldsymbol{\theta}_{PS}$ in Theorem 3 is given by*

$$\mathbb{E}\left\|\boldsymbol{\theta}_\infty - \boldsymbol{\theta}_{PS}\right\|^2 = \mathcal{O}\left(\frac{G^2}{\tilde{\mu}^2}\left(1 + \frac{d \log(1/\delta)}{m^2 \varepsilon^2}\right)\right). \quad (16)$$

*which is achieved by setting the clipping threshold as*

$$c^\star = \frac{2Gm^2\varepsilon^2}{d \log(1/\delta) + 2m^2\varepsilon^2}.$$

We observe that the asymptotic deviation from $\boldsymbol{\theta}_{PS}$ is in the order of $\mathcal{O}(d/(\tilde{\mu}^2\varepsilon^2))$ which contains the combined effects from the sensitivity of distribution shift and DP requirement.

# 4. Reducing Clipping Bias in Clipped SGD

The previous section illustrates that **PCSGD** suffers from a *bias amplification* phenomenon in the setting of performative prediction. Although the issue can be remedied by

---

**Algorithm 1 DiceSGD** with Greedy Deployment

1: **Input:** $C_1, C_2, a_0, a_1, \varepsilon, \delta, \mathsf{D}_0, \sigma_{\mathsf{DP}}^2$ with $C_2 \geq C_1$, initialization $\boldsymbol{\theta}_0, e_0 = 0$.
2: **for** $t = 0$ to $T - 1$ **do**
3: $\quad$ Draw new sample $Z_{t+1} \sim \mathcal{D}(\boldsymbol{\theta}_t)$ and Gaussian noise $\zeta_{t+1} \sim \mathcal{N}(0, \sigma_{\mathsf{DP}}^2 \mathbf{I})$.
4: $\quad v_{t+1} = \mathsf{clip}_{C_1}(\nabla\ell(\boldsymbol{\theta}_t; Z_{t+1})) + \mathsf{clip}_{C_2}(e_t)$.
5: $\quad \boldsymbol{\theta}_{t+1} = \boldsymbol{\theta}_t - \gamma_{t+1}(v_{t+1} + \zeta_{t+1})$,
$\quad\quad e_{t+1} = e_t + \nabla\ell(\boldsymbol{\theta}_t; Z_{t+1}) - v_{t+1}$.
6: **end for**
7: **Output:** Last iterate $\boldsymbol{\theta}_T$.

---

tuning the step size $\gamma$ and clipping threshold $c$, as in Corollaries 2, 3, it may not be feasible for practical applications as the problem parameters such as $\tilde{\mu}, G$ may be unknown.

In this section, we discuss how to reduce the clipping bias through applying a recently proposed clipped SGD algorithm **DiceSGD** (Zhang et al., 2024). The latter is proven to achieve DP while converging to the exact solution of a convex optimization problem. We adapt this algorithm in the performative prediction setting and show that it removes the clipping bias inflicted by **PCSGD**.

For the subsequent discussion, we consider the unconstrained setting $\mathcal{X} \equiv \mathbb{R}^d$. The **DiceSGD** algorithm is summarized in Algorithm 1 with the greedy deployment mechanism for the performative prediction setting. Compared to **PCSGD**, the notable differences include the use of *two* clipping operators in line 4 for forming the stochastic gradient estimate $v_{t+1}$, and an *error feedback* step in line 5 where $e_t$ accumulates the error due to clipping. The vector $e_t \in \mathbb{R}^d$ is a private variable kept by the learner.

We remark that the pseudo code describes a general implementation which includes the Gaussian noise mechanism for privacy protection. As shown in (Zhang et al., 2024), the use of two clipping operators reduces the privacy leakage. With an appropriate $\sigma_{\mathsf{DP}}^2$, the algorithm is guaranteed to achieve $(\varepsilon, \delta)$-DP through reparameterization of the notion of Renyi DP, a relaxed notion for DP proposed in (Mironov, 2017). Nonetheless, when the DP requirement is not needed, one may set $\sigma_{\mathsf{DP}}^2 = 0$ for a clipped algorithm with reduced bias.

Importantly, the error feedback mechanism is effective in removing the asymptotic bias. To see that this insight can be extended to the performative prediction setting, observe that any fixed point $(\bar{e}, \bar{\boldsymbol{\theta}})$ of Algorithm 1 satisfies

$$-\mathsf{clip}_{C_2}(\bar{e}) = \mathbb{E}_{Z \sim \mathcal{D}(\bar{\boldsymbol{\theta}})}[\mathsf{clip}_{C_1}(\nabla\ell(\bar{\boldsymbol{\theta}}; Z))]$$
$$\nabla f(\bar{\boldsymbol{\theta}}; \bar{\boldsymbol{\theta}}) - \mathsf{clip}_{C_2}(\bar{e}) = \mathbb{E}_{Z \sim \mathcal{D}(\bar{\boldsymbol{\theta}})}[\mathsf{clip}_{C_1}(\nabla\ell(\bar{\boldsymbol{\theta}}; Z))] \quad (17)$$

Under the condition $C_2 \geq C_1$, a feasible fixed point $(\bar{e}, \bar{\boldsymbol{\theta}})$ shall satisfy $\nabla f(\bar{\boldsymbol{\theta}}; \bar{\boldsymbol{\theta}}) = \mathbf{0}$ and $\bar{e} =$

$-\mathbb{E}_{Z\sim\mathcal{D}(\bar{\boldsymbol{\theta}})}[\mathsf{clip}_{C_1}(\nabla\ell(\bar{\boldsymbol{\theta}};Z))]$ since

$$\|\bar{e}\| \le \mathbb{E}_{Z\sim\mathcal{D}(\bar{\boldsymbol{\theta}})}[\|\mathsf{clip}_{C_1}(\nabla\ell(\bar{\boldsymbol{\theta}};Z))\|] \le C_1, \qquad (18)$$

where the first inequality is due to Jensen's inequality. The condition $\nabla f(\bar{\boldsymbol{\theta}};\bar{\boldsymbol{\theta}})=\mathbf{0}$ implies $\bar{\boldsymbol{\theta}}=\boldsymbol{\theta}_{PS}$ under strongly-convex $\ell(\cdot)$.

We conclude by presenting the convergence for the **DiceSGD** algorithm in the performative prediction setting. We first observe the assumption:

**A8.** *There exists a constant $M$ such that for any $t \ge 1$, $\mathbb{E}[\|e_t\|^2] \le M^2$.*

A8 is verified empirically in our experiments for the case of $C_2 \ge C_1$; see §H. Similar to **PCSGD**, the **DiceSGD** algorithm is also a non-gradient algorithm with non-smooth drifts. The following analysis is achieved by designing a suitable Lyapunov function for each type of $\ell(\cdot)$.

**Strongly Convex Loss**  Notice that A4 implies that there exists $G, B \ge 0$ with

$$\mathbb{E}_{Z\sim\mathcal{D}(\boldsymbol{\theta})}[\|\nabla\ell(\boldsymbol{\theta};Z)\|^2] \le G^2 + B^2\|\boldsymbol{\theta}-\boldsymbol{\theta}_{PS}\|^2, \quad (19)$$

for any $\boldsymbol{\theta} \in \mathbb{R}^d$. Denote $\tilde{\mu} := \mu - L\beta$ and $\overline{\boldsymbol{\theta}}_t := \boldsymbol{\theta}_t - \gamma_t e_t - \boldsymbol{\theta}_{PS}$, we have

**Theorem 7.** *Under A1, 2, 4, 5, 8 and (19) holds. Suppose that $\beta < \frac{\mu}{L}$, there exists $a_0, a_1, b, \bar{b} \ge 0$ such that the step size of **DiceSGD** satisfies i) $\gamma_t = a_0/(a_1+t)$, $a_0 \ge 1/b$, ii) $\gamma_t \le \min\{\tilde{\mu}/(16b), 8/\tilde{\mu}, \tilde{\mu}/4B^2\}$, and iii) $\gamma_t^2/\gamma_{t+1}^2 \le 1 + \bar{b}\gamma_{t+1}^2$, Then, for any $t \ge 0$,*

$$\mathbb{E}\left\|\overline{\boldsymbol{\theta}}_{t+1}\right\|^2 \le \prod_{i=1}^{t+1}\left(1 - \frac{\tilde{\mu}}{4}\gamma_i\right)\|\overline{\boldsymbol{\theta}}_0\|^2 + \frac{8(G^2+d\sigma_{\mathsf{DP}}^2)}{\tilde{\mu}}\gamma_{t+1}$$
$$+ \frac{16L^2M^2(1+\beta)^2}{\tilde{\mu}^2}\gamma_{t+1}^2 + \frac{24b^2M^2}{\tilde{\mu}}\gamma_{t+1}^3$$
$$+ \frac{16L^2M^2\bar{b}(1+\beta)^2}{\tilde{\mu}^2}\gamma_{t+1}^4. \qquad (20)$$

We present the detailed proof in §F. Under strong convex losses A1, our analysis shows that **DiceSGD** converges to a unique fixed point ($\boldsymbol{\theta}_{PS}$) in the mean square sense. Note that the analysis framework here differs significantly from that for non-convex losses.

We observe from (20) that the dominant term on the right-hand-side is the second term which behaves as $\mathcal{O}(\gamma_{t+1}) = \mathcal{O}(1/t)$. Consequently, we have

$$\mathbb{E}\|\boldsymbol{\theta}_t - \boldsymbol{\theta}_{PS}\|^2 \le 2\mathbb{E}\left\|\overline{\boldsymbol{\theta}}_t\right\|^2 + 2\gamma_t^2\mathbb{E}\|e_t\|^2 = \mathcal{O}(1/t),$$

due to A8. In other words, the **DiceSGD** algorithm is asymptotically *unbiased*.

**Non-convex Loss**  Our last endeavor is to show that for the non-convex loss setting, **DiceSGD** also reduces the bias due to clipping in performative prediction settings. We observe the convergence result:

**Theorem 8.** *Under A2, 4, 6, 7, 8. If we set $\gamma = 1/\sqrt{T}$, then for sufficiently large $T$, it holds*

$$\min_{t=0,\ldots,T-1}\mathbb{E}[\|\nabla f(\boldsymbol{\theta}_t;\boldsymbol{\theta}_t)\|^2] = \mathcal{O}\left(\frac{1}{\sqrt{T}} + \mathsf{b}\beta\right), \quad (21)$$

*where $\mathsf{b} = \mathcal{O}(\ell_{\max}((C_1+C_2) + \sqrt{d}\sigma_{\mathsf{DP}}))$.*

The detailed theorem and proof can be found in §G. Our analysis involved a few modifications over (Zhang et al., 2024, Theorem 3.6) for the decision-dependent distribution.

Unlike the convergence Theorems 1 & 5 for the **PCSGD** algorithm, the analysis of **DiceSGD** relaxes the uniformly bounded gradient assumption (A3) to the variance-based assumption (A4). This relaxation is enabled by the feedback mechanism embedded in the **DiceSGD** algorithm.

Importantly, from (21), we observe that when $\beta \approx 0$, there is no asymptotic bias for **DiceSGD**. However, we also note that unlike **PCSGD**, the multiplicative factor b depends on $\sigma_{\mathsf{DP}}^2, C_1, C_2$ which indicates that the bias due to distribution shift may become more sensitive when using **DiceSGD**.

**Remark 2.** *The DP guarantees of **DiceSGD** with distribution shift can be studied by extending (Zhang et al., 2024, Theorem 3.7 & Appendix A.2). Particularly, we can model the distribution shift through a random mapping $f : (Z + D_0) \mapsto D_0$, similar to the one introduced in Appendix E. Applying the data processing inequality shows that a comparable bound for the DP guarantees of the **DiceSGD** algorithm under distribution shift to (Zhang et al., 2024, Theorem 3.7) can be established.*

## 5. Numerical Experiments

All experiments are performed with Python on a server using a single Intel Xeon 6138 CPU thread. In the interest of space, we only consider the experiments with strongly convex $\ell(\cdot)$ and focus on the setting with DP guarantees to validate our theoretical findings. For the sake of fair comparison between the **PCSGD** and the **DiceSGD** algorithms, we choose the set $\mathcal{X}$ to be such that the optimal point of the unconstrained and constrained case of (2) will coincide. To maintain the same DP guarantees, we respectively set the DP noise standard deviation for **PCSGD** and **DiceSGD** as $\sigma_{\mathsf{DP}}$ and $\sqrt{96}\sigma_{\mathsf{DP}}$, according to Corollary 1 and (Zhang et al., 2024, Theorem 3.7). For the **DiceSGD** algorithm, we set $C_1 = C_2$. This decision is motivated by the necessity to maintain a balanced trade-off in Algorithm 1, where augmenting the values of $C_1$ and $C_2$ would entail an increase in the variance of the Gaussian mechanism in line 3.

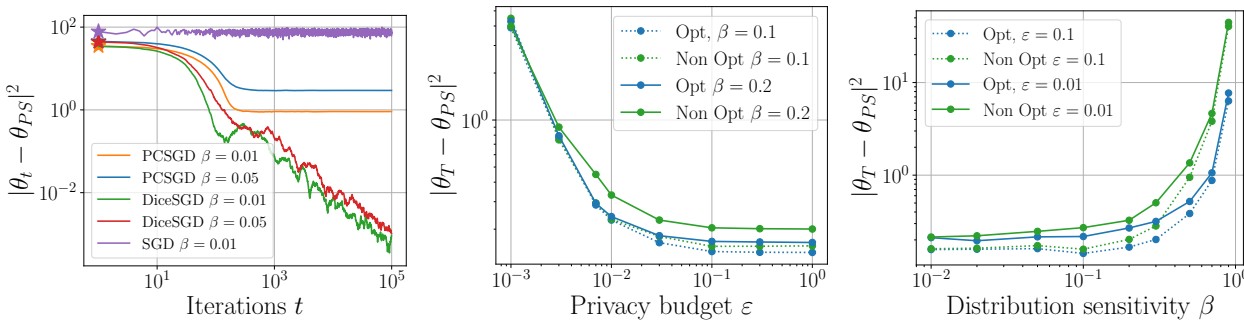

Figure 1. **Quadratic Minimization** (First) The performative stability gap $\|\boldsymbol{\theta}_t - \boldsymbol{\theta}_{PS}\|^2$. (Second) Trade off between privacy budget $\varepsilon$ and bias. (Third) Bias amplification effect due to $\beta$.

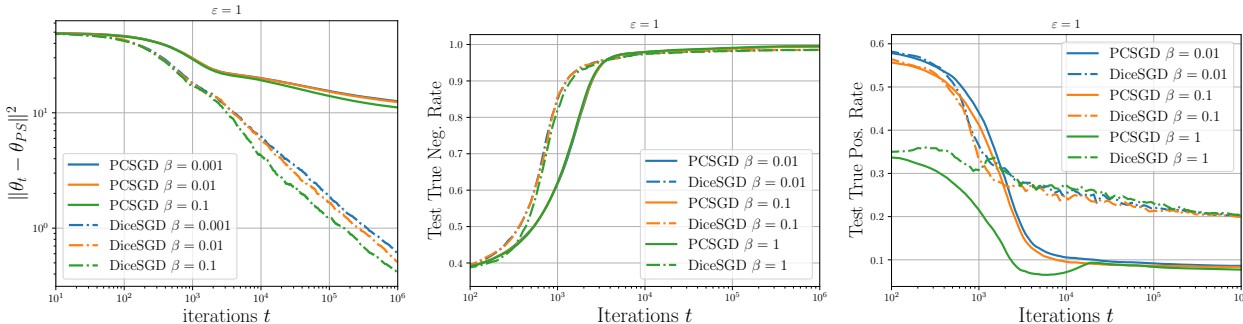

Figure 2. **Logistic Regression** (First) Gap between iterations and performative stable point $\|\boldsymbol{\theta}_t - \boldsymbol{\theta}_{PS}\|^2$. (Second) Test true negative rate with shifted distribution. (Third) Test true positive accuracy with shifted distribution.

We remark that additional experiment details and results can be found in Appendix H, including an example for the case with non-convex loss in Appendix H.3.

**Quadratic Minimization.** The *first problem* is concerned with the validation of Theorems 3, 4, and 7. Here, we consider a scalar performative risk optimization problem with synthetic data

$$\min_{\boldsymbol{\theta} \in \mathcal{X}} \mathbb{E}_{z \sim \mathcal{D}(\boldsymbol{\theta})}[(\boldsymbol{\theta} + az)^2/2],$$

where $\mathcal{D}(\boldsymbol{\theta})$ is a uniform distribution over the data points $\{b\tilde{Z}_i - \beta\boldsymbol{\theta}\}_{i=1}^m$ such that $\tilde{Z}_i \sim \mathcal{B}(p)$ is Bernoulli and $a > 0, b > 0, p < 1/2$. We also set $\mathcal{X} = [-10, 10]$ and observe that for $0 < \beta < a^{-1}$, the performative stable solution is $\boldsymbol{\theta}_{PS} = -\frac{\bar{p}a}{1-a\beta}$, where $\bar{p} = \frac{1}{m}\sum_{i=1}^m \tilde{Z}_i$ is the sample mean.

We set $p = 0.1, \varepsilon = 0.1, \delta = 1/m, \beta \in \{0.01, 0.05\}, a = 10, b = 1, c = C_1 = C_2 = 1$, the sample size $m = 10^5$. The step size is $\gamma_t = \frac{10}{100+t}$ with the initialization $\boldsymbol{\theta}_0 = 5$. In Fig. 1 (first plot), we compare $|\boldsymbol{\theta}_t - \boldsymbol{\theta}_{PS}|^2$ against the iteration number $t$ using plain **SGD** with DP noise, **PCSGD** and **DiceSGD**. As observed, adding the DP noise compromises **SGD**'s convergence. **PCSGD** cannot converge to $\boldsymbol{\theta}_{PS}$ due to the clipping bias which increases as $\beta \uparrow$. Meanwhile, **DiceSGD** finds a *bias-free* solution as it converges to $\boldsymbol{\theta}_{PS}$ at rate $\mathcal{O}(1/t)$.

Our next experiments examine the trade-off between clipping bias of **PCSGD** $|\boldsymbol{\theta}_T - \boldsymbol{\theta}_{PS}|^2$ and privacy budget $\varepsilon$ or distribution sensitivity $\beta$. We set $a = 1, b = 6, c = c^\star \approx 2.32, T = 10^5, \beta \in \{0.1, 0.2\}$ or $\varepsilon \in \{0.01, 0.1\}$, while keeping the other parameters unchanged. Using Corollary 2, we set the optimal step size according to $\gamma^\star$ in (15), and the non-optimal step size as $\gamma = \frac{\log(1/\Delta(\mu))}{\mu T}$ to simulate the scenario when the presence of distribution shift is unknown. From Fig. 1 (second & third plots), setting the optimal step size $\gamma^\star$ adapted to distribution shifts achieves a smaller bias in all settings. Meanwhile, as the privacy budget decreases $\varepsilon \downarrow 0$ or the sensitivity of distribution shift increases $\beta \uparrow \frac{\mu}{L}$, the bias of **PCSGD** increases.

**Logistic Regression.** We consider the real dataset `GiveMeSomeCredit` (Kaggle, 2011) with $m = 15776$ samples and $d = 10$ features. We split the training/test sets using the ratio of $7 : 3$. The learner aims to find a classifier via minimizing the regularized logistic loss:

$$\ell(\boldsymbol{\theta}; z) = \alpha(z)\left(\log(1 + \exp(x^\top\boldsymbol{\theta})) - yx^\top\boldsymbol{\theta}\right) + \frac{\eta}{2}\|\boldsymbol{\theta}\|^2,$$

where $\eta = 10^2/m$ is a regularization parameter, $z \equiv (x, y) \in \mathbb{R}^d \times \{0, 1\}$ is the training sample, $\alpha(z) = y + 1$ is a label weight, $y = 0$ $(y = 1)$ denotes a customer without (with) history of defaults. The strategic behavior of the population, i.e., their features $x$ are adapted to $\boldsymbol{\theta}$ through

maximizing a quadratic utility function.

We fix the privacy budget at $\varepsilon = 1$, $\delta = 1/m$, clipping thresholds $c = C_1 = C_2 = 1$, step size $\gamma_t = 50/(5000+t)$, and sensitivity parameter $\beta \in \{0.001, 0.01, 0.1\}$. From Fig. 2 (first plot), we observe the gap $\|\boldsymbol{\theta}_t - \boldsymbol{\theta}_{PS}\|^2$ of **DiceSGD** decays at $\mathcal{O}(1/t)$ as $t \to \infty$, **PCSGD** achieve the steady state due to bias after the transient stage. This coincides with Theorems 3 & 7. In Fig. 2 (second & third plots), we compare the test accuracy against the iteration number $t$. Here, the trajectory becomes more unstable as $\beta \uparrow$. An interesting observation is that increasing the sensitivity $\beta$ leads to lower true positive rate.

## 6. Conclusions

This paper initiates the study of clipped SGD algorithms in the performative prediction setting. In both cases with strongly convex and non-convex losses, we discovered a *bias amplification* effect with the **PCSGD** algorithm and proposed several remedies including an extension of the **DiceSGD** algorithm to performative prediction.

## Impact Statement

This paper presents work whose goal is to advance the field of Machine Learning. There are many potential societal consequences of our work, none which we feel must be specifically highlighted here.

## Acknowledgments

This work was supported by the Shun Hing Institute of Advanced Engineering, The Chinese University of Hong Kong, under Project #MMT-p5-23.

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

## A. Useful Facts

In the proof of Theorems 3 & 7, we utilize the following Lemma which is introduced in (Li et al., 2022, Lemma 6). The lemma upper bounds the coefficients of contraction equations in the form of (28) and (45):

**Lemma 1.** *(Li et al., 2022, Lemma 6) Consider a sequence of non-negative, non-increasing step sizes $\{\gamma_t\}_{t\geq 1}$. Let $a > 0$, $p \in \mathbb{Z}_+$ and $\gamma_1 < 2/a$. If*

$$\left(\frac{\gamma_t}{\gamma_{t+1}}\right)^p \leq 1 + \frac{a}{2} \cdot \gamma_{t+1}^p$$

*for any $t \geq 1$, then*

$$\sum_{j=1}^{t} \gamma_j^{p+1} \prod_{\ell=j+1}^{t} (1 - \gamma_\ell a) \leq \frac{2}{a} \gamma_t^p, \quad \forall\, t \geq 1. \tag{22}$$

The proof of this lemma is presented in (Li et al., 2022, Lemma 6) and is therefore omitted.

Additionally, we rely on the following lemma that provides smoothness guarantees regarding the performative prediction gradients:

**Lemma 2.** *Under A2, 5. For any $\boldsymbol{\theta}_0, \boldsymbol{\theta}_1, \boldsymbol{\theta}, \boldsymbol{\theta}' \in \mathbb{R}^d$, it holds that*

$$\|\nabla f(\boldsymbol{\theta}_0, \boldsymbol{\theta}) - \nabla f(\boldsymbol{\theta}_1, \boldsymbol{\theta}')\| \leq L \|\boldsymbol{\theta}_0 - \boldsymbol{\theta}_1\| + L\beta \|\boldsymbol{\theta} - \boldsymbol{\theta}'\|. \tag{23}$$

The proof of this lemma can be found in (Drusvyatskiy & Xiao, 2023, Lemma 2.1).

## B. Proof of Theorem 3

We outline the main steps in proving the convergence for **PCSGD**. In particular, we shall consider the general form of **PCSGD** in (14) where the DP noise is introduced. To this end, we aim at proving the following bound:

**Theorem.** *Under A1, 2, 3, 5. Suppose that $\beta < \frac{\mu}{L}$, the step sizes $\{\gamma_t\}_{t\geq 1}$ are non-increasing and satisfy i) $\frac{\gamma_{t-1}}{\gamma_t} \leq 1 + \frac{\mu - L\beta}{2}\gamma_t$, and ii) $\gamma_t \leq \frac{2}{\mu - L\beta}$. Then, for any $t \geq 1$, the expected squared distance between $\boldsymbol{\theta}_t$ and the performative stable solution $\boldsymbol{\theta}_{PS}$ satisfies*

$$\mathbb{E}\|\hat{\boldsymbol{\theta}}_{t+1}\|^2 \leq \prod_{i=1}^{t+1}(1 - \tilde{\mu}\gamma_i)\|\hat{\boldsymbol{\theta}}_0\|^2 + \frac{2c_1}{\tilde{\mu}}\gamma_{t+1} + \frac{8\mathcal{C}_1}{\tilde{\mu}^2},$$

*where $c_1 := 2(c^2 + G^2) + d\sigma_{\mathsf{DP}}^2$, $\mathcal{C}_1 := (\max\{G - c, 0\})^2$, and $\tilde{\mu} := \mu - L\beta$.*

To simplify notations, we define

$$\widetilde{\nabla} g(\boldsymbol{\theta}_t) := \mathsf{clip}_c(\nabla \ell(\boldsymbol{\theta}_t; Z_{t+1})), \quad b_t := \widetilde{\nabla} g(\boldsymbol{\theta}_t) - \nabla f(\boldsymbol{\theta}_t; \boldsymbol{\theta}_{PS}). \tag{24}$$

Recall that $\hat{\boldsymbol{\theta}}_t := \boldsymbol{\theta}_t - \boldsymbol{\theta}_{PS}$, the following lemma characterizes the one-step progress of **PCSGD**.

**Lemma 3.** *Under A2, 3, 5. For any $t \geq 0$, it holds*

$$\mathbb{E}_t\|\hat{\boldsymbol{\theta}}_{t+1}\|^2 \leq (1 - 2\mu\gamma_{t+1})\|\hat{\boldsymbol{\theta}}_t\|^2 + \gamma_{t+1}^2\big(\min\{c^2, G^2\} + d\sigma_{\mathsf{DP}}^2\big) - 2\gamma_{t+1}\left\langle \hat{\boldsymbol{\theta}}_t \mid \mathbb{E}_t[b_t]\right\rangle. \tag{25}$$

The proof is in §B.1. The inner product term above involving $\mathbb{E}_t[b_t]$ captures the clipping bias and the distribution shift. The latter is the difference between the clipped stochastic gradient and the expected gradient induced by $\mathcal{D}(\boldsymbol{\theta}_{PS})$.

Such term is unlikely to be small except for a large clipping threshold $c$. For example, (Zhang et al., 2020b, Lemma 9) applied an indicator function trick to bound $\|\mathbb{E}_t[b_t]\|$ by $G/c$. We improve their treatment on the bias term via a better use of the smoothness property (cf. Lemma 2) to obtain:

**Lemma 4.** *Under A2, 3, 5, the following upper bound holds:*

$$-2\gamma_{t+1}\langle\boldsymbol{\theta}_t - \boldsymbol{\theta}_{PS}\,|\,\mathbb{E}_t[b_t]\rangle \le 2L\beta\gamma_{t+1}\|\boldsymbol{\theta}_t - \boldsymbol{\theta}_{PS}\|^2 + \tilde{\mu}\gamma_{t+1}\|\boldsymbol{\theta}_t - \boldsymbol{\theta}_{PS}\|^2 + \frac{4\gamma_{t+1}}{\tilde{\mu}}\cdot\mathcal{C}_1,\tag{26}$$

*with the constants* $\mathcal{C}_1 := (\max\{G - c, 0\})^2$, $\tilde{\mu} := \mu - L\beta$.

See §B.2 for the detailed proof. The first two terms on the right hand side of Eq. (26) vanishes as $\|\boldsymbol{\theta}_t - \boldsymbol{\theta}_{PS}\| \to 0$. Meanwhile, the term $\frac{4\gamma_{t+1}\mathcal{C}_1}{\tilde{\mu}}$ has led to an inevitable clipping bias. Notice that this term vanishes only in the trivial case of $c \ge G$, i.e., the clipping threshold is larger than any stochastic gradient. Otherwise, this bias will propagate through the algorithm and lead to an asymptotic bias.

**Proof of Theorem 3.** Combining Lemmas 3, 4 leads to the following recursion:

$$\mathbb{E}_t\|\hat{\boldsymbol{\theta}}_{t+1}\|^2 \le (1 - \tilde{\mu}\gamma_{t+1})\|\hat{\boldsymbol{\theta}}_t\|^2 + \frac{4\mathcal{C}_1}{\tilde{\mu}}\gamma_{t+1} + \gamma_{t+1}^2\left(2c^2 + 2G^2 + d\sigma_{\mathsf{DP}}^2\right).\tag{27}$$

Recall that $c_1 := (2c^2 + 2G^2 + d\sigma_{\mathsf{DP}}^2)$. Taking full expectation on both sides of (27) leads to:

$$\mathbb{E}\|\hat{\boldsymbol{\theta}}_{t+1}\|^2 \le \prod_{i=1}^{t+1}(1 - \tilde{\mu}\gamma_i)\left\|\hat{\boldsymbol{\theta}}_0\right\|^2 + c_1\sum_{i=1}^{t+1}\gamma_i^2\prod_{j=i+1}^{t+1}(1 - \tilde{\mu}\gamma_j) + \frac{4\mathcal{C}_1}{\tilde{\mu}}\sum_{i=1}^{t+1}\gamma_i\prod_{j=i+1}^{t+1}(1 - \tilde{\mu}\gamma_j)$$

$$\le \prod_{i=1}^{t+1}(1 - \tilde{\mu}\gamma_i)\left\|\hat{\boldsymbol{\theta}}_0\right\|^2 + \frac{8\mathcal{C}_1}{\tilde{\mu}^2} + \frac{2c_1}{\tilde{\mu}}\gamma_{t+1},\tag{28}$$

where we used Lemma 1 in the last inequality for step size satisfying $\sup_{t\ge 1}\gamma_t \le \frac{2}{\tilde{\mu}}$. $\qquad\square$

## B.1. Proof of Lemma 3

*Proof.* Recall the notation $\widetilde{\nabla}g(\boldsymbol{\theta}_t) := \mathsf{clip}_c(\nabla\ell(\boldsymbol{\theta}_t; Z_{t+1}))$ which we introduce in (24). Then, we deduce the following chain

$$\|\boldsymbol{\theta}_{t+1} - \boldsymbol{\theta}_{PS}\|^2 \overset{(a)}{=} \left\|\mathcal{P}_{\mathcal{X}}\left(\boldsymbol{\theta}_t - \gamma_{t+1}\left[\widetilde{\nabla}g(\boldsymbol{\theta}_t) + \zeta_{t+1}\right]\right) - \mathcal{P}_{\mathcal{X}}\left(\boldsymbol{\theta}_{PS} + \gamma_{t+1}\nabla f(\boldsymbol{\theta}_{PS}; \boldsymbol{\theta}_{PS})\right)\right\|^2$$

$$\overset{(b)}{\le} \left\|\boldsymbol{\theta}_t - \gamma_{t+1}\left[\widetilde{\nabla}g(\boldsymbol{\theta}_t) + \zeta_{t+1}\right] - \boldsymbol{\theta}_{PS} + \gamma_{t+1}\nabla f(\boldsymbol{\theta}_{PS}; \boldsymbol{\theta}_{PS})\right\|^2$$

$$= \|\boldsymbol{\theta}_t - \boldsymbol{\theta}_{PS}\|^2 + \gamma_{t+1}^2\left\|\widetilde{\nabla}g(\boldsymbol{\theta}_t) + \zeta_{t+1} - \nabla f(\boldsymbol{\theta}_{PS}; \boldsymbol{\theta}_{PS})\right\|^2$$

$$- 2\gamma_{t+1}\left\langle\boldsymbol{\theta}_t - \boldsymbol{\theta}_{PS}\,|\,\widetilde{\nabla}g(\boldsymbol{\theta}_t) + \zeta_{t+1} - \nabla f(\boldsymbol{\theta}_{PS}; \boldsymbol{\theta}_{PS})\right\rangle,$$

where in equality $(a)$, we applied the definition of $\boldsymbol{\theta}_{PS}$ in (2). Inequality $(b)$ is due to the non-expansive property of the projection operator. Introducing notation $b_t := \widetilde{\nabla}g(\boldsymbol{\theta}_t) - \nabla f(\boldsymbol{\theta}_t; \boldsymbol{\theta}_{PS})$ which we define in (24) into the above inequality gives us

$$\|\boldsymbol{\theta}_{t+1} - \boldsymbol{\theta}_{PS}\|^2 \le \|\boldsymbol{\theta}_t - \boldsymbol{\theta}_{PS}\|^2 + \gamma_{t+1}^2\left\|\widetilde{\nabla}g(\boldsymbol{\theta}_t) + \zeta_{t+1} - \nabla f(\boldsymbol{\theta}_{PS}; \boldsymbol{\theta}_{PS})\right\|^2$$

$$- 2\gamma_{t+1}\langle\boldsymbol{\theta}_t - \boldsymbol{\theta}_{PS}\,|\,b_t + \zeta_{t+1} + \nabla f(\boldsymbol{\theta}_t; \boldsymbol{\theta}_{PS}) - \nabla f(\boldsymbol{\theta}_{PS}; \boldsymbol{\theta}_{PS})\rangle$$

$$\overset{(a)}{\le} (1 - 2\gamma_{t+1}\mu)\|\boldsymbol{\theta}_t - \boldsymbol{\theta}_{PS}\|^2 + \gamma_{t+1}^2\left\|\widetilde{\nabla}g(\boldsymbol{\theta}_t) + \zeta_{t+1} - \nabla f(\boldsymbol{\theta}_{PS}; \boldsymbol{\theta}_{PS})\right\|^2$$

$$- 2\gamma_{t+1}\langle\boldsymbol{\theta}_t - \boldsymbol{\theta}_{PS}\,|\,b_t + \zeta_{t+1}\rangle,$$

where inequality $(a)$ is due to strong convexity of $\ell(\cdot; z)$, i.e., $\langle\boldsymbol{\theta}_t - \boldsymbol{\theta}_{PS}\,|\,\nabla f(\boldsymbol{\theta}_t; \boldsymbol{\theta}_{PS}) - \nabla f(\boldsymbol{\theta}_{PS}; \boldsymbol{\theta}_{PS})\rangle \ge \mu\|\boldsymbol{\theta}_t - \boldsymbol{\theta}_{PS}\|^2$. Taking conditional expectation with regards to $\boldsymbol{\theta}_t$ on both sides gives us

$$\mathbb{E}_t\|\boldsymbol{\theta}_{t+1} - \boldsymbol{\theta}_{PS}\|^2 \le (1 - 2\mu\gamma_{t+1})\|\boldsymbol{\theta}_t - \boldsymbol{\theta}_{PS}\|^2 - 2\gamma_{t+1}\langle\boldsymbol{\theta}_t - \boldsymbol{\theta}_{PS}\,|\,\mathbb{E}_t[b_t]\rangle$$

$$+ \gamma_{t+1}^2\mathbb{E}_t\left\|\widetilde{\nabla}g(\boldsymbol{\theta}_t) + \zeta_{t+1} - \nabla f(\boldsymbol{\theta}_{PS}; \boldsymbol{\theta}_{PS})\right\|^2.\tag{29}$$

For the last term of right-hand-side of the above inequality, we have

$$\mathbb{E}_t \left\| \widetilde{\nabla} g(\boldsymbol{\theta}_t) + \zeta_{t+1} - \nabla f(\boldsymbol{\theta}_{PS}; \boldsymbol{\theta}_{PS}) \right\|^2$$

$$= \mathbb{E}_t \left[ \| \widetilde{\nabla} g(\boldsymbol{\theta}_t) - \nabla f(\boldsymbol{\theta}_{PS}; \boldsymbol{\theta}_{PS}) \|^2 + \langle \widetilde{\nabla} g(\boldsymbol{\theta}_t) - \nabla f(\boldsymbol{\theta}_{PS}; \boldsymbol{\theta}_{PS}) \,|\, \zeta_{t+1} \rangle + \| \zeta_{t+1} \|^2 \right]$$

$$\overset{(a)}{=} \mathbb{E}_t \| \widetilde{\nabla} g(\boldsymbol{\theta}_t) - \nabla f(\boldsymbol{\theta}_{PS}; \boldsymbol{\theta}_{PS}) \|^2 + 0 + d\sigma_{\mathsf{DP}}^2$$

$$= \mathbb{E}_t \| \mathsf{clip}_c(\nabla \ell(\boldsymbol{\theta}_t; z_{t+1})) - \nabla f(\boldsymbol{\theta}_{PS}; \boldsymbol{\theta}_{PS}) \|^2 + d\sigma_{\mathsf{DP}}^2$$

$$\leq 2\mathbb{E}_t \left\| \mathsf{clip}_c(\nabla \ell(\boldsymbol{\theta}_t; z_{t+1})) \right\|^2 + 2 \left\| \nabla f(\boldsymbol{\theta}_{PS}; \boldsymbol{\theta}_{PS}) \right\|^2 + d\sigma_{\mathsf{DP}}^2 \overset{(b)}{\leq} 2(c^2 + G^2) + d\sigma_{\mathsf{DP}}^2,$$

where $(a)$ is obtained since the additive perturbing noise is zero mean and statistically independent of the stochastic gradient, $(b)$ is due to A 3 and definition of clipping operator.

Substituting above upper bound to (29) leads to

$$\mathbb{E}_t \left\| \boldsymbol{\theta}_{t+1} - \boldsymbol{\theta}_{PS} \right\|^2 \leq (1 - 2\mu\gamma_{t+1}) \| \boldsymbol{\theta}_t - \boldsymbol{\theta}_{PS} \|^2 - 2\gamma_{t+1} \langle \boldsymbol{\theta}_t - \boldsymbol{\theta}_{PS} \,|\, \mathbb{E}_t[b_t] \rangle$$
$$+ \gamma_{t+1}^2 \left( 2c^2 + 2G^2 + d\sigma_{\mathsf{DP}}^2 \right),$$

which finishes the proof of Lemma 3. $\qquad\square$

### B.2. Proof of Lemma 4

*Proof.* Applying the Cauchy-Schwarz inequality on the inner product $\langle \boldsymbol{\theta}_{PS} - \boldsymbol{\theta}_t \,|\, \mathbb{E}_t[b_t] \rangle$ yields the following upper bound:

$$-2\gamma_{t+1} \langle \boldsymbol{\theta}_t - \boldsymbol{\theta}_{PS} \,|\, \mathbb{E}_t[b_t] \rangle \leq 2\gamma_{t+1} \| \boldsymbol{\theta}_t - \boldsymbol{\theta}_{PS} \| \cdot \| \mathbb{E}_t[b_t] \| .$$

We now proceed to upper bounding $\| \mathbb{E}_t[b_t] \|$:

$$\| \mathbb{E}_t[b_t] \| = \left\| \mathbb{E}_t \left[ \widetilde{\nabla} g(\boldsymbol{\theta}_t) - \nabla f(\boldsymbol{\theta}_t; \boldsymbol{\theta}_{PS}) \right] \right\|$$

$$= \| \nabla g(\boldsymbol{\theta}_t; \boldsymbol{\theta}_t) - \nabla f(\boldsymbol{\theta}_t; \boldsymbol{\theta}_{PS}) \|$$

$$\leq \| \nabla g(\boldsymbol{\theta}_t; \boldsymbol{\theta}_t) - \nabla f(\boldsymbol{\theta}_t; \boldsymbol{\theta}_t) \| + \| \nabla f(\boldsymbol{\theta}_t; \boldsymbol{\theta}_t) - \nabla f(\boldsymbol{\theta}_t; \boldsymbol{\theta}_{PS}) \|$$

$$\leq \left\| \mathbb{E}_{z \sim \mathcal{D}(\boldsymbol{\theta}_t)} \left( \mathsf{clip}_c(\nabla \ell(\boldsymbol{\theta}_t; z)) - \nabla \ell(\boldsymbol{\theta}_t; z) \right) \right\| + L\beta \| \boldsymbol{\theta}_t - \boldsymbol{\theta}_{PS} \|$$

where the last inequality is due to Assumption 2 and Lemma D.4 of (Perdomo et al., 2020). Moreover,

$$\left\| \mathbb{E}_{z \sim \mathcal{D}(\boldsymbol{\theta}_t)} \left[ \min \left( 1, \frac{c}{\| \nabla \ell(\boldsymbol{\theta}_t; z) \|} \right) - 1 \right] \nabla \ell(\boldsymbol{\theta}_t; z) \right\|$$

$$\leq \mathbb{E}_{z \sim \mathcal{D}(\boldsymbol{\theta}_t)} \left[ \left| 1 - \min \left( 1, \frac{c}{\| \nabla \ell(\boldsymbol{\theta}_t; z) \|} \right) \right| \cdot \| \nabla \ell(\boldsymbol{\theta}_t; z) \| \right] \qquad (30)$$

$$= \mathbb{E}_{z \sim \mathcal{D}(\boldsymbol{\theta}_t)} \left[ \max \left( 0, \| \nabla \ell(\boldsymbol{\theta}_t; z) \| - c \right) \right] \leq \max\{G - c, 0\},$$

where we have used Assumption 3 in the last inequality. Finally, we obtain

$$-2\gamma_{t+1} \langle \boldsymbol{\theta}_t - \boldsymbol{\theta}_{PS} \,|\, \mathbb{E}_t[b_t] \rangle \leq 2\gamma_{t+1} \| \boldsymbol{\theta}_t - \boldsymbol{\theta}_{PS} \| \cdot (\max\{G - c, 0\} + L\beta \| \boldsymbol{\theta}_t - \boldsymbol{\theta}_{PS} \|)$$

$$= 2L\beta\gamma_{t+1} \| \boldsymbol{\theta}_t - \boldsymbol{\theta}_{PS} \|^2 + 2\gamma_t \| \boldsymbol{\theta}_t - \boldsymbol{\theta}_{PS} \| \cdot \max\{G - c, 0\}$$

$$\leq 2L\beta\gamma_{t+1} \| \boldsymbol{\theta}_t - \boldsymbol{\theta}_{PS} \|^2 + \tilde{\mu}\gamma_{t+1} \| \boldsymbol{\theta}_t - \boldsymbol{\theta}_{PS} \|^2 + \frac{4\gamma_{t+1}}{\tilde{\mu}} \cdot (\max\{G - c, 0\})^2,$$

where in the last inequality, we used the Hölder's inequality $xy \leq ax^2 + \frac{y^2}{a}$ and set $a = \tilde{\mu}/2$. $\qquad\square$

## C. Proof of Theorem 4

*Proof.* Let $a, b, \beta > 0$ and denote by $\mathcal{B}(p)$ the Bernoulli distribution with mean $0 < p < \frac{1}{2}$. We consider the following quadratic loss function:

$$\min_{\boldsymbol{\theta} \in \mathcal{X}} \mathbb{E}_{Z \sim \mathcal{D}(\boldsymbol{\theta})} [\ell(\boldsymbol{\theta}; Z)],$$

$$\text{where } \ell(\boldsymbol{\theta}; z) = \tfrac{1}{2}(\boldsymbol{\theta} + az)^2 \text{ and } Z \sim \mathcal{D}(\boldsymbol{\theta}) \iff Z = b\tilde{Z} - \beta\boldsymbol{\theta}, \ \tilde{Z} \sim \mathcal{B}(p). \tag{31}$$

We require that $ab \geq 2c$. When $Z \sim \mathcal{D}(\boldsymbol{\theta})$, we note that the stochastic gradient is in the form:

$$\nabla\ell(\boldsymbol{\theta}; Z) = \boldsymbol{\theta} + a(b\tilde{Z} - \beta\boldsymbol{\theta}) = (1 - a\beta)\boldsymbol{\theta} + ab\tilde{Z}.$$

Problem (31) satisfies A2 with $\mu = 1$, $L = a$ and A5 with the parameter $\beta$.

Consider the following point:

$$\boldsymbol{\theta}_\infty = -\frac{pc}{(1 - p)(1 - a\beta)}.$$

We claim that $\boldsymbol{\theta}_\infty$ is a fixed point of the clipped SGD algorithm. Note that the stochastic gradient at $\boldsymbol{\theta}_\infty$ with $Z \sim \mathcal{D}(\boldsymbol{\theta}_\infty)$ when $\tilde{Z} = 0$ is not clipped, since

$$\left|(1 - a\beta)\boldsymbol{\theta}_\infty + ab \cdot 0\right| = \left|(1 - a\beta) \times \left(-\frac{pc}{(1 - p)(1 - a\beta)}\right)\right| = \left|-\frac{pc}{1 - p}\right| \leq c,$$

where the last inequality is due to the assumption $p < \frac{1}{2}$. Meanwhile, when $\tilde{Z} = 1$, the stochastic gradient is:

$$(1 - a\beta)\boldsymbol{\theta}_\infty + ab = -\frac{pc}{1 - p} + ab \geq -c + 2c = c,$$

where the last inequality uses the condition $ab \geq 2c$. In particular, we have that:

$$\mathbb{E}_{Z \sim \mathcal{D}(\boldsymbol{\theta})}[\text{clip}_c(\nabla\ell(\boldsymbol{\theta}_\infty; Z))] = (1 - p) \cdot [(1 - a\beta)\boldsymbol{\theta}] + pc = 0.$$

The above shows that $\boldsymbol{\theta}_\infty$ is a fixed point of the Clipped SGD algorithm.

On the other hand, the performative stable solution, $\boldsymbol{\theta}_{PS}$, of problem (31) solves the following equation,

$$\mathbb{E}_{z \sim \mathcal{D}(\boldsymbol{\theta}_{PS})}[\nabla\ell(\boldsymbol{\theta}_{PS}; z)] = \mathbb{E}_{Z \sim \mathcal{D}(\boldsymbol{\theta}_{PS})}(\boldsymbol{\theta}_{PS} + aZ) = \boldsymbol{\theta}_{PS} + a(p - \beta\boldsymbol{\theta}_{PS}) = 0,$$

we get that

$$\boldsymbol{\theta}_{PS} = -\frac{pa}{1 - a\beta}.$$

Finally, we can lower bound the asymptotic bias of clipped SGD as

$$\|\boldsymbol{\theta}_\infty - \boldsymbol{\theta}_{PS}\|^2 = \left\|-\frac{pc}{(1 - p)(1 - a\beta)} + \frac{pa}{1 - a\beta}\right\|^2 = \frac{p^2}{(1 - a\beta)^2}\left(a - \frac{c}{1 - p}\right)^2 = \Omega\left(\frac{1}{(\mu - L\beta)^2}\right),$$

Finally, we aim to prove that $\boldsymbol{\theta}_\infty$ is unique. We remark that the limiting points of clipping SGD satisfies

$$\mathbb{E}_{z \sim \mathcal{D}(\boldsymbol{\theta})}\text{clip}_c(\nabla\ell(\boldsymbol{\theta}; z)) = 0,$$

which is equivalent to

$$p \times \text{clip}_c[(1 - a\beta)\boldsymbol{\theta} + ab] + (1 - p) \times \text{clip}_c[(1 - a\beta)\boldsymbol{\theta}] = 0. \tag{32}$$

**Case 1**: $(1 - a\beta)\boldsymbol{\theta} + ab > c$ and $(1 - a\beta)\boldsymbol{\theta} < c$, then, we will get $\boldsymbol{\theta}_\infty$.

**Case 2**: $(1 - a\beta)\boldsymbol{\theta} + ab > c$ and $(1 - a\beta)\boldsymbol{\theta} > c$, then

$$p \times c + (1 - p) \times c = 1 \neq 0.$$

**Case 3**: $(1 - a\beta)\boldsymbol{\theta} + ab < c$ and $(1 - a\beta)\boldsymbol{\theta} < c$, i.e, it requires that $\boldsymbol{\theta} < \frac{c - ab}{1 - a\beta}$, then from (32), we obtain that

$$p \times [(1 - a\beta)\boldsymbol{\theta} + ab] + (1 - p) \times [(1 - a\beta)\boldsymbol{\theta}] = 0.$$

Solve it, we get $\boldsymbol{\theta} = \frac{pab}{1 - a\beta}$. Note that

$$\boldsymbol{\theta} = \frac{pab}{1 - a\beta} \leq \frac{c - ab}{1 - a\beta},$$

which is impossible, since $ab \geq 2c$ and $p, a > 0$. In conclusion, there is only one solution of (32). This concludes the proof. $\qquad\square$

# D. Proof of Theorem 5

In this section, we consider a special case of (1) where $\mathcal{X} = \mathbb{R}^d$. Similar to §B, we also consider the general setting of **PCSGD** with DP noise (14).

**Theorem.** *Under A2, 3, 4, 6, 7. Let the step sizes satisfy* $\sup_{t \geq 1} \gamma_t \leq \frac{1}{2(1+\sigma_1^2)}$. *Then, for any* $T \geq 1$, *the iterates* $\{\boldsymbol{\theta}_t\}_{t \geq 0}$ *generates by* (4) *satisfy:*

$$\sum_{t=0}^{T-1} \gamma_{t+1} \mathbb{E}\left[\|\nabla f(\boldsymbol{\theta}_t; \boldsymbol{\theta}_t)\|^2\right] \leq 8\Delta_0 + 4L(\sigma_0^2 + \sigma_{\mathsf{DP}}^2) \sum_{t=0}^{T-1} \gamma_{t+1}^2 + 8\mathsf{b}(\beta, c) \sum_{t=0}^{T-1} \gamma_{t+1},$$

*where* $\Delta_0 := \mathbb{E}[f(\boldsymbol{\theta}_0; \boldsymbol{\theta}_0) - \ell^\star]$ *is an upper bound to the initial optimality gap for performative risk, and*

$$\mathsf{b}(\beta, c) := \widehat{L}\beta \left( \sqrt{\sigma_0^2 + \sigma_{\mathsf{DP}}^2} + 8(1+\sigma_1^2)\widehat{L}\beta \right) + 2\max\{G-c, 0\}^2,$$

*where* $\widehat{L} := \ell_{\mathsf{max}}$.

*Proof of Theorem 5 with general* $\sigma_{\mathsf{DP}}^2$. For fixed $z \in \mathsf{Z}$, applying A2 leads to

$$\ell(\boldsymbol{\theta}_{t+1}; z) \leq \ell(\boldsymbol{\theta}_t; z) + \langle \nabla \ell(\boldsymbol{\theta}_t; z) \,|\, \boldsymbol{\theta}_{t+1} - \boldsymbol{\theta}_t \rangle + \frac{L}{2}\|\boldsymbol{\theta}_{t+1} - \boldsymbol{\theta}_t\|^2$$

$$= \ell(\boldsymbol{\theta}_t; z) - \gamma_{t+1}\langle \nabla\ell(\boldsymbol{\theta}_t; z) \,|\, \mathsf{clip}_c\left(\nabla\ell(\boldsymbol{\theta}_t; z_{t+1})\right) + \zeta_{t+1} \rangle + \frac{L\gamma_{t+1}^2}{2}\left\|\mathsf{clip}_c(\nabla\ell(\boldsymbol{\theta}_{t;z_{t+1}})) + \zeta_{t+1}\right\|^2$$

$$= \ell(\boldsymbol{\theta}_t; z) - \gamma_{t+1}\langle \nabla\ell(\boldsymbol{\theta}_t; z) \,|\, \widetilde{\nabla}g(\boldsymbol{\theta}_t) + \zeta_{t+1} \rangle + \frac{L\gamma_{t+1}^2}{2}\left\|\widetilde{\nabla}g(\boldsymbol{\theta}_t) + \zeta_{t+1}\right\|^2.$$

The second line is due to the update rule of (4). In the last line, we recall the notation $\widetilde{\nabla}g(\boldsymbol{\theta}_t) := \mathsf{clip}_c(\nabla\ell(\boldsymbol{\theta}_t; z_{t+1}))$. Taking integration on fixed $z$ with weights given by the p.d.f. of $\mathcal{D}(\boldsymbol{\theta}_t)$, i.e., $\int (\cdot) p_{\boldsymbol{\theta}_t}(z) dz$ on both sides of above inequality yield

$$f(\boldsymbol{\theta}_{t+1}; \boldsymbol{\theta}_t) \leq f(\boldsymbol{\theta}_t; \boldsymbol{\theta}_t) - \gamma_{t+1}\langle \nabla f(\boldsymbol{\theta}_t; \boldsymbol{\theta}_t) \,|\, \widetilde{\nabla}g(\boldsymbol{\theta}_t) + \zeta_{t+1} \rangle + \frac{L\gamma_{t+1}^2}{2}\left\|\widetilde{\nabla}g(\boldsymbol{\theta}_t) + \zeta_{t+1}\right\|^2.$$

Taking conditional expectation $\mathbb{E}_t[\cdot]$ on the clipping stochastic gradients leads to

$$f(\boldsymbol{\theta}_{t+1}; \boldsymbol{\theta}_t) \leq f(\boldsymbol{\theta}_t; \boldsymbol{\theta}_t) - \gamma_{t+1}\langle \nabla f(\boldsymbol{\theta}_t; \boldsymbol{\theta}_t) \,|\, \mathbb{E}_t\left[\widetilde{\nabla}g(\boldsymbol{\theta}_t)\right] \rangle + \frac{L\gamma_{t+1}^2}{2}\mathbb{E}_t\left[\left\|\widetilde{\nabla}g(\boldsymbol{\theta}_t) + \zeta_{t+1}\right\|^2\right]. \tag{33}$$

where we use the property that $\zeta_{t+1} \sim \mathcal{N}(0, \sigma_{\mathsf{DP}}^2 \boldsymbol{I})$. For the last term in above, we observe the following chain,

$$\mathbb{E}_t\left[\left\|\widetilde{\nabla}g(\boldsymbol{\theta}_t) + \zeta_{t+1}\right\|^2\right] = \mathbb{E}_t\left\|\widetilde{\nabla}g(\boldsymbol{\theta}_t)\right\|^2 + 0 + \sigma_{\mathsf{DP}}^2 \overset{(a)}{\leq} \mathbb{E}_t\|\nabla\ell(\boldsymbol{\theta}_t; z_{t+1})\|^2 + \sigma_{\mathsf{DP}}^2$$

$$= \mathbb{E}_t\|\nabla\ell(\boldsymbol{\theta}_t; z_{t+1}) - \nabla f(\boldsymbol{\theta}_t; \boldsymbol{\theta}_t)\|^2 + \|\nabla f(\boldsymbol{\theta}_t; \boldsymbol{\theta}_t)\|^2 + \sigma_{\mathsf{DP}}^2$$

$$\overset{(b)}{\leq} \sigma_0^2 + \sigma_{\mathsf{DP}}^2 + (1+\sigma_1^2)\|\nabla f(\boldsymbol{\theta}_t; \boldsymbol{\theta}_t)\|^2,$$

where $(a)$ used the definition of clipping operator (6) and $(b)$ used A4. Substituting above results to (33) gives us

$$f(\boldsymbol{\theta}_{t+1}; \boldsymbol{\theta}_t) \leq f(\boldsymbol{\theta}_t; \boldsymbol{\theta}_t) - \gamma_{t+1}\left\langle \nabla f(\boldsymbol{\theta}_t; \boldsymbol{\theta}_t) \,\middle|\, \mathbb{E}_t\left[\widetilde{\nabla}g(\boldsymbol{\theta}_t)\right] \right\rangle + \frac{L\gamma_{t+1}^2}{2}\left(\sigma_0^2 + \sigma_{\mathsf{DP}}^2 + (1+\sigma_1^2)\|\nabla f(\boldsymbol{\theta}_t; \boldsymbol{\theta}_t)\|^2\right). \tag{34}$$

The inner product term can be lower bounded by

$$\left\langle \nabla f(\boldsymbol{\theta}_t; \boldsymbol{\theta}_t) \,\middle|\, \mathbb{E}_t\left[\widetilde{\nabla}g(\boldsymbol{\theta}_t)\right] \right\rangle = \|\nabla f(\boldsymbol{\theta}_t; \boldsymbol{\theta}_t)\|^2 - \langle \nabla f(\boldsymbol{\theta}_t; \boldsymbol{\theta}_t) \,|\, \nabla g(\boldsymbol{\theta}_t) - \nabla f(\boldsymbol{\theta}_t; \boldsymbol{\theta}_t) \rangle$$

$$\geq \|\nabla f(\boldsymbol{\theta}_t; \boldsymbol{\theta}_t)\|^2 - \|\nabla f(\boldsymbol{\theta}_t; \boldsymbol{\theta}_t)\| \cdot \|\nabla g(\boldsymbol{\theta}_t) - \nabla f(\boldsymbol{\theta}_t; \boldsymbol{\theta}_t)\|$$

$$\overset{(a)}{\geq} \|\nabla f(\boldsymbol{\theta}_t; \boldsymbol{\theta}_t)\|^2 - \max\{G - c, 0\} \|\nabla f(\boldsymbol{\theta}_t; \boldsymbol{\theta}_t)\|$$

$$\geq \frac{1}{2} \|\nabla f(\boldsymbol{\theta}_t; \boldsymbol{\theta}_t)\|^2 - 2\max\{G - c, 0\}^2,$$

where $(a)$ is due to (30) in Lemma 4 and the last inequality is due to the fact that $xy \leq ax^2 + \frac{y^2}{a}$, for any $a > 0$. Substituting back to (34) gives

$$f(\boldsymbol{\theta}_{t+1}; \boldsymbol{\theta}_t) \leq f(\boldsymbol{\theta}_t; \boldsymbol{\theta}_t) - \frac{1}{2} \|\nabla f(\boldsymbol{\theta}_t; \boldsymbol{\theta}_t)\|^2 + 2\max\{G - c, 0\}^2 + \frac{L\gamma_{t+1}^2}{2} \left( \sigma_0^2 + \sigma_{\mathsf{DP}}^2 + (1 + \sigma_1^2) \|\nabla f(\boldsymbol{\theta}_t; \boldsymbol{\theta}_t)\|^2 \right).$$

Rearranging terms,

$$\left( \frac{1}{2} - \frac{L(1 + \sigma_1^2)\gamma_{t+1}}{2} \right) \|\nabla f(\boldsymbol{\theta}_t; \boldsymbol{\theta}_t)\|^2 \leq f(\boldsymbol{\theta}_t; \boldsymbol{\theta}_t) - f(\boldsymbol{\theta}_{t+1}; \boldsymbol{\theta}_t) + 2\max\{G - c, 0\}^2 + \frac{L(\sigma_0^2 + \sigma_{\mathsf{DP}}^2)}{2} \gamma_{t+1}^2.$$

The step size condition $\sup_{t \geq 1} \gamma_t \leq \frac{1}{2(1 + \sigma_1^2)}$ implies that $\frac{1}{2} - \frac{L(1 + \sigma_1^2)\gamma_{t+1}}{2} \geq \frac{1}{4}$.

$$\frac{1}{4} \|\nabla f(\boldsymbol{\theta}_t; \boldsymbol{\theta}_t)\|^2 \leq f(\boldsymbol{\theta}_t; \boldsymbol{\theta}_t) - f(\boldsymbol{\theta}_{t+1}; \boldsymbol{\theta}_t) + 2\max\{G - c, 0\}^2 + \frac{L(\sigma_0^2 + \sigma_{\mathsf{DP}}^2)}{2} \gamma_{t+1}^2. \tag{35}$$

Next, taking full expectation on both sides and decomposing the term $f(\boldsymbol{\theta}_t; \boldsymbol{\theta}_t) - f(\boldsymbol{\theta}_{t+1}; \boldsymbol{\theta}_t)$ as:

$$\mathbb{E}[f(\boldsymbol{\theta}_t; \boldsymbol{\theta}_t) - f(\boldsymbol{\theta}_{t+1}; \boldsymbol{\theta}_t)] = \mathbb{E}[f(\boldsymbol{\theta}_t; \boldsymbol{\theta}_t) - f(\boldsymbol{\theta}_{t+1}; \boldsymbol{\theta}_{t+1})] + \mathbb{E}[f(\boldsymbol{\theta}_{t+1}; \boldsymbol{\theta}_{t+1}) - f(\boldsymbol{\theta}_{t+1}; \boldsymbol{\theta}_t)]$$

$$\overset{(a)}{\leq} \mathbb{E}[f(\boldsymbol{\theta}_t; \boldsymbol{\theta}_t) - f(\boldsymbol{\theta}_{t+1}; \boldsymbol{\theta}_{t+1})] + \widehat{L}\beta \|\boldsymbol{\theta}_{t+1} - \boldsymbol{\theta}_t\|$$

$$\overset{(b)}{\leq} \mathbb{E}[f(\boldsymbol{\theta}_t; \boldsymbol{\theta}_t) - f(\boldsymbol{\theta}_{t+1}; \boldsymbol{\theta}_{t+1})] + \widehat{L}\beta\gamma_{t+1}\mathbb{E}_t \left\| \widetilde{\nabla} g(\boldsymbol{\theta}_t) + \zeta_{t+1} \right\|$$

$$\overset{(c)}{\leq} \mathbb{E}[f(\boldsymbol{\theta}_t; \boldsymbol{\theta}_t) - f(\boldsymbol{\theta}_{t+1}; \boldsymbol{\theta}_{t+1})] + \widehat{L}\beta\gamma_{t+1}\sqrt{\mathbb{E}_t \left\| \widetilde{\nabla} g(\boldsymbol{\theta}_t) + \zeta_{t+1} \right\|^2} \tag{36}$$

$$\overset{(d)}{\leq} \mathbb{E}[f(\boldsymbol{\theta}_t; \boldsymbol{\theta}_t) - f(\boldsymbol{\theta}_{t+1}; \boldsymbol{\theta}_{t+1})] + \widehat{L}\beta\gamma_{t+1} \left( \sqrt{\sigma_0^2 + \sigma_{\mathsf{DP}}^2} + \sqrt{(\sigma_1^2 + 1)} \|\nabla f(\boldsymbol{\theta}_t; \boldsymbol{\theta}_t)\| \right)$$

$$\leq \mathbb{E}[f(\boldsymbol{\theta}_t; \boldsymbol{\theta}_t) - f(\boldsymbol{\theta}_{t+1}; \boldsymbol{\theta}_{t+1})]$$

$$+ \widehat{L}\beta\gamma_{t+1} \left( \sqrt{\sigma_0^2 + \sigma_{\mathsf{DP}}^2} + (1 + \sigma_1^2) \cdot 8\widehat{L}\beta + \frac{1}{8\widehat{L}\beta} \|\nabla f(\boldsymbol{\theta}_t; \boldsymbol{\theta}_t)\|^2 \right).$$

Inequality $(a)$ is due to (Li & Wai, 2024, Lemma 3), $(b)$ is due to the update rule (4) and $(c)$ is implied by $[\mathbb{E}X]^2 \leq \mathbb{E}[X^2]$. In $(d)$, we apply A4. Back to (35), we have

$$\frac{1}{8}\gamma_{t+1}\mathbb{E} \|\nabla f(\boldsymbol{\theta}_t; \boldsymbol{\theta}_t)\|^2 \leq \mathbb{E}\left[ f(\boldsymbol{\theta}_t; \boldsymbol{\theta}_t) - f(\boldsymbol{\theta}_{t+1}; \boldsymbol{\theta}_{t+1}) \right] + \frac{L(\sigma_0^2 + \sigma_{\mathsf{DP}}^2)}{2}\gamma_{t+1}^2 + \mathsf{b}(\beta, c) \cdot \gamma_{t+1},$$

where $\mathsf{b}(\beta, c)$ is a constant depending on distribution shifting strength and clipping threshold, which defined as

$$\mathsf{b}(\beta, c) := \widehat{L}\beta \left( \sqrt{\sigma_0^2 + \sigma_{\mathsf{DP}}^2} + 8(1 + \sigma_1^2)\widehat{L}\beta \right) + 2\max\{G - c, 0\}^2. \tag{37}$$

Taking summation from $t = 0, 1, \cdots, T - 1$ gives us

$$\frac{1}{8} \sum_{t=0}^{T-1} \gamma_{t+1}\mathbb{E} \|\nabla f(\boldsymbol{\theta}_t; \boldsymbol{\theta}_t)\|^2 \leq \Delta_0 + \frac{L(\sigma_0^2 + \sigma_{\mathsf{DP}}^2)}{2} \sum_{t=0}^{T-1} \gamma_{t+1}^2 + \mathsf{b}(\beta, c) \cdot \sum_{t=0}^{T-1} \gamma_{t+1}.$$

Set $\gamma_t = 1/\sqrt{T}$ and divide $\sum_{t=0}^{T-1} \gamma_{t+1}$ on both sides yields the theorem. $\qquad \square$

# E. Proof of Corollary 1

*Proof.* Let $o$ and $aux$ denote an outcome and an auxiliary input, respectively. We wish to prove

$$\Pr(\mathcal{M}(aux, \boldsymbol{Z}) = o) \leq e^\varepsilon \Pr(\mathcal{M}(aux, \boldsymbol{Z}') = o) + \delta,$$

where $\boldsymbol{Z}$ and $\boldsymbol{Z}'$ are two adjacent datasets, i.e., they are only different by one sample. Let us define the privacy loss of an outcome $o$ on two datasets $\boldsymbol{Z}$ and $\boldsymbol{Z}'$ as

$$c(o; \mathcal{M}, aux, \boldsymbol{Z}, \boldsymbol{Z}') := \log \frac{\Pr(\mathcal{M}(aux, \boldsymbol{Z}) = o)}{\Pr(\mathcal{M}(aux, \boldsymbol{Z}') = o)}$$

and its log moment generating function as

$$\alpha^{\mathcal{M}}(\lambda; aux, \boldsymbol{Z}, \boldsymbol{Z}') := \log \mathbb{E}_{o \sim \mathcal{M}(aux, \boldsymbol{Z})}[\exp(\lambda \cdot c(o; \mathcal{M}, aux, \boldsymbol{Z}, \boldsymbol{Z}'))],$$

where $\mathcal{M}$ is the mechanism we focus on, $\lambda > 0$ is the variable of log moment generating function. Taking maximum over conditions, the unconditioned log moment generating function is

$$\hat{\alpha}^{\mathcal{M}}(\lambda) := \max_{aux, \boldsymbol{Z}, \boldsymbol{Z}'} \alpha^{\mathcal{M}}(\lambda; aux, \boldsymbol{Z}, \boldsymbol{Z}').$$

The overall log moment generating function can be bounded as following according to composability in (Abadi et al., 2016, Theorem 2)

$$\hat{\alpha}^{\mathcal{M}}(\lambda) \leq \sum_{t=1}^{T} \hat{\alpha}^{\mathcal{M}^{(t)}}(\lambda).$$

Let $q = \frac{1}{m}$ denotes the probability each data sample is drawn uniformly from the dataset $\mathsf{D}_0$. Recall (13), which leads to $\bar{z}_i = Z - s_i(\theta)$ where $i$ is chosen uniformly from $[m]$ and $s_i(\theta)$ is generated according to the random mapping $\mathcal{S}_i : \mathcal{X} \to \mathsf{Z}$. We can define this relationship by the random mapping $f : (\mathsf{Z} + \mathsf{D}_0) \to \mathsf{D}_0$, where for $X \subseteq \mathbb{R}^d$ and $Y \subseteq \mathbb{R}^d$ we let $X + Y$ denote the set with all the variables $\{x + y : x \in X, y \in Y\}$ and the randomness follows from the uniform distribution of $i \in [m]$ and the random mapping $\mathcal{S}_i : \mathcal{X} \to \mathsf{Z}$. By (Dwork & Roth, 2014, Proposition 2.1) using the post processing function $f$, after the privacy mechanism $\mathcal{M}$, i.e. $f \circ \mathcal{M}$, does not compromise differential privacy.

Observe that the distribution shift due to $\mathcal{S}_i(\boldsymbol{\theta})$ in (13) cannot compromise differential privacy (Dwork & Roth, 2014, Proposition 2.1). Let $q \leq \frac{c}{16\sigma_{\mathsf{DP}}}$ and $\lambda \leq \frac{\sigma_{\mathsf{DP}}^2}{c^2} \log \frac{c}{q\sigma_{\mathsf{DP}}}$, we apply (Abadi et al., 2016, Lemma 3) to bound the unconditioned log moment generating function as

$$\hat{\alpha}^{\mathcal{M}^{(t)}}(\lambda) \leq \frac{q^2 \lambda(1 + \lambda)c^2}{(1 - q)\sigma_{\mathsf{DP}}^2} + \mathcal{O}\left(\frac{q^3 \lambda^3 c^3}{\sigma_{\mathsf{DP}}^3}\right) = \mathcal{O}\left(\frac{q^2 \lambda^2 c^2}{\sigma_{\mathsf{DP}}^2}\right).$$

Finally, we only need to verify that there exists some $\lambda$ that satisfies the following inequalities

$$T\left(\frac{qc\lambda}{\sigma_{\mathsf{DP}}}\right)^2 \leq \frac{\lambda\varepsilon}{2}, \quad \exp(-\lambda\varepsilon/2) \leq \delta, \quad \lambda \leq \frac{\sigma_{\mathsf{DP}}^2}{c^2} \log\left(\frac{c}{q\sigma_{\mathsf{DP}}}\right).$$

We can verify that when $\varepsilon = c_1 q^2 T$, $q = 1/m$ and $\sigma_{\mathsf{DP}} = \frac{cq\sqrt{T \log(1/\delta)}}{\varepsilon}$, all above conditions can be satisfied for some explicit constant $c_1$. This finishes the proof. $\square$

# F. Proof of Theorem 7

For the convenience of derivations, we introduce the following notations: $\widetilde{\boldsymbol{\theta}}_t = \boldsymbol{\theta}_t - \gamma_t e_t$. Its update rule is

$$
\begin{aligned}
\widetilde{\boldsymbol{\theta}}_{t+1} &= \boldsymbol{\theta}_{t+1} - \gamma_{t+1} e_{t+1} \\
&= \boldsymbol{\theta}_t - \gamma_{t+1}(v_{t+1} + \zeta_t) - \gamma_t\left(e_t + \nabla\ell(\boldsymbol{\theta}_t; z_t) - v_{t+1}\right) \qquad (38) \\
&= \boldsymbol{\theta}_t - \gamma_{t+1}\zeta_{t+1} - \gamma_{t+1}\left(e_t + \nabla\ell(\boldsymbol{\theta}_t; z_{t+1})\right)
\end{aligned}
$$

$$= \quad \widetilde{\boldsymbol{\theta}}_t + (\gamma_t - \gamma_{t+1}) e_t - \gamma_{t+1} (\zeta_{t+1} + \nabla\ell(\boldsymbol{\theta}_t; z_{t+1})).$$

We remark that if we choose a constant step size, i.e, $\gamma_t \equiv \gamma$, then $\widetilde{\boldsymbol{\theta}}_{t+1} = \widetilde{\boldsymbol{\theta}}_t - \gamma (\zeta_{t+1} + \nabla\ell(\boldsymbol{\theta}_t; z_{t+1}))$, which degenerates to the case in the proof of (Zhang et al., 2024).

Meanwhile, we can obtain the feedback error $e_t$'s update rule:

$$
\begin{aligned}
e_{t+1} &= e_t + \nabla\ell(\boldsymbol{\theta}_t; z_{t+1}) - \mathsf{clip}_{C_2}(e_t) - \mathsf{clip}_{C_1}(\nabla\ell(\boldsymbol{\theta}_t; z_{t+1})) \\
&= (1 - \alpha_e^t)e_t + (1 - \alpha^t)\nabla\ell(\boldsymbol{\theta}_t; z_{t+1}),
\end{aligned}
\tag{39}
$$

where we define $\alpha_e^t, \alpha^t$ as following:

$$
\alpha_e^t := \min\left\{1, \frac{C_1}{\|e_t\|}\right\}, \quad \alpha^t := \min\left\{1, \frac{C_2}{\|\nabla\ell(\boldsymbol{\theta}_t; z_{t+1})\|}\right\}.
\tag{40}
$$

We aim to analyze the squared distance between $\widetilde{\boldsymbol{\theta}}_{t+1}$ and $\boldsymbol{\theta}_{PS}$.

$$
\begin{aligned}
\left\|\widetilde{\boldsymbol{\theta}}_{t+1} - \boldsymbol{\theta}_{PS}\right\|^2 &= \left\|\widetilde{\boldsymbol{\theta}}_t + (\gamma_t - \gamma_{t+1})e_t - \gamma_{t+1}(\nabla\ell(\boldsymbol{\theta}_t; z_{t+1}) + \zeta_{t+1}) - \boldsymbol{\theta}_{PS}\right\|^2 \\
&= \left\|\widetilde{\boldsymbol{\theta}}_t - \boldsymbol{\theta}_{PS}\right\|^2 + \|(\gamma_t - \gamma_{t+1})e_t - \gamma_{t+1}[\nabla\ell(\boldsymbol{\theta}_t; z_{t+1}) + \zeta_{t+1}]\|^2 \\
&\quad + 2\left\langle \widetilde{\boldsymbol{\theta}}_t - \boldsymbol{\theta}_{PS} \,|\, (\gamma_t - \gamma_{t+1})e_t - \gamma_{t+1}[\nabla\ell(\boldsymbol{\theta}_t; z_{t+1}) + \zeta_{t+1}]\right\rangle.
\end{aligned}
$$

Tacking conditional expectation $\mathbb{E}_t[\cdot]$ on both sides leads to

$$
\begin{aligned}
\mathbb{E}_t\left\|\widetilde{\boldsymbol{\theta}}_{t+1} - \boldsymbol{\theta}_{PS}\right\|^2 &\leq \left\|\widetilde{\boldsymbol{\theta}}_t - \boldsymbol{\theta}_{PS}\right\|^2 + 2\|(\gamma_t - \gamma_{t+1})e_t\|^2 \\
&\quad + 2\gamma_{t+1}^2\left[\mathbb{E}_t\|\nabla\ell(\boldsymbol{\theta}_t; z_{t+1})\|^2 + d\sigma_{\mathsf{DP}}^2\right] + 2\left\langle \widetilde{\boldsymbol{\theta}}_t - \boldsymbol{\theta}_{PS} \,|\, (\gamma_t - \gamma_{t+1})e_t\right\rangle \\
&\quad - 2\gamma_{t+1}\left\langle \widetilde{\boldsymbol{\theta}}_t - \boldsymbol{\theta}_{PS} \,|\, \nabla f(\boldsymbol{\theta}_t; \boldsymbol{\theta}_t) - \nabla f(\boldsymbol{\theta}_{PS}, \boldsymbol{\theta}_{PS})\right\rangle,
\end{aligned}
\tag{41}
$$

where we use inequality $(a + b)^2 \leq 2a^2 + 2b^2$, $\zeta_{t+1} \sim \mathcal{N}(0, \sigma_{\mathsf{DP}}^2\mathbf{I})$ and the fact that $\nabla f(\boldsymbol{\theta}_{PS}, \boldsymbol{\theta}_{PS}) = 0$.

Applying (19) leads to following upper bound:

$$
\begin{aligned}
\mathbb{E}_t\|\widetilde{\boldsymbol{\theta}}_{t+1} - \boldsymbol{\theta}_{PS}\|^2 &\leq (1 + 2\gamma_{t+1}^2 B^2)\left\|\widetilde{\boldsymbol{\theta}}_t - \boldsymbol{\theta}_{PS}\right\|^2 + 2(\gamma_t - \gamma_{t+1})^2\|e_t\|^2 \\
&\quad + 2\gamma_{t+1}^2\left[G^2 + d\sigma_{\mathsf{DP}}^2\right] + \underbrace{2\left\langle \widetilde{\boldsymbol{\theta}}_t - \boldsymbol{\theta}_{PS} \,|\, (\gamma_t - \gamma_{t+1})e_t\right\rangle}_{:=B_1} \\
&\quad - 2\gamma_{t+1}\underbrace{\left\langle \widetilde{\boldsymbol{\theta}}_t - \boldsymbol{\theta}_{PS} \,|\, \nabla f(\boldsymbol{\theta}_t; \boldsymbol{\theta}_t) - \nabla f(\boldsymbol{\theta}_{PS}, \boldsymbol{\theta}_{PS})\right\rangle}_{:=B_2}.
\end{aligned}
\tag{42}
$$

Now, let's consider the term $B_1$ first. Using Holder's inequality, we have

$$
B_1 \leq 4(\gamma_t - \gamma_{t+1})\left[\left\|\widetilde{\boldsymbol{\theta}}_t - \boldsymbol{\theta}_{PS}\right\|^2 + \|e_t\|^2\right].
$$

For the term $B_2$, we observe the following chain,

$$
\begin{aligned}
B_2 &= \left\langle \widetilde{\boldsymbol{\theta}}_t - \boldsymbol{\theta}_{PS} \,|\, \nabla f(\boldsymbol{\theta}_t; \boldsymbol{\theta}_t) - \nabla f(\widetilde{\boldsymbol{\theta}}_t; \boldsymbol{\theta}_{PS}) + \nabla f(\widetilde{\boldsymbol{\theta}}_t; \boldsymbol{\theta}_{PS}) - \nabla f(\boldsymbol{\theta}_{PS}; \boldsymbol{\theta}_{PS})\right\rangle \\
&\stackrel{(a)}{\geq} \mu\left\|\widetilde{\boldsymbol{\theta}}_t - \boldsymbol{\theta}_{PS}\right\|^2 + \left\langle \widetilde{\boldsymbol{\theta}}_t - \boldsymbol{\theta}_{PS} \,|\, \nabla f(\boldsymbol{\theta}_t; \boldsymbol{\theta}_t) - \nabla f(\widetilde{\boldsymbol{\theta}}_t; \boldsymbol{\theta}_{PS})\right\rangle \\
&= \mu\left\|\widetilde{\boldsymbol{\theta}}_t - \boldsymbol{\theta}_{PS}\right\|^2 - \left\langle \boldsymbol{\theta}_{PS} - \widetilde{\boldsymbol{\theta}}_t \,|\, \nabla f(\boldsymbol{\theta}_t; \boldsymbol{\theta}_t) - \nabla f(\widetilde{\boldsymbol{\theta}}_t; \widetilde{\boldsymbol{\theta}}_t)\right\rangle - \left\langle \boldsymbol{\theta}_{PS} - \widetilde{\boldsymbol{\theta}}_t \,|\, \nabla f(\widetilde{\boldsymbol{\theta}}_t; \widetilde{\boldsymbol{\theta}}_t) - \nabla f(\widetilde{\boldsymbol{\theta}}_t; \boldsymbol{\theta}_{PS})\right\rangle
\end{aligned}
$$

$$\overset{(b)}{\geq} \mu \left\| \widetilde{\boldsymbol{\theta}}_t - \boldsymbol{\theta}_{PS} \right\|^2 - \left\| \widetilde{\boldsymbol{\theta}}_t - \boldsymbol{\theta}_{PS} \right\| \cdot L(1+\beta) \left\| \boldsymbol{\theta}_t - \widetilde{\boldsymbol{\theta}}_t \right\| - L\beta \left\| \widetilde{\boldsymbol{\theta}}_t - \boldsymbol{\theta}_{PS} \right\|^2$$

$$\overset{(c)}{=} \tilde{\mu} \left\| \widetilde{\boldsymbol{\theta}}_t - \boldsymbol{\theta}_{PS} \right\|^2 - L(1+\beta) \left\| \widetilde{\boldsymbol{\theta}}_t - \boldsymbol{\theta}_{PS} \right\| \cdot \| \gamma_t e_t \|$$

$$\geq \tilde{\mu} \left\| \widetilde{\boldsymbol{\theta}}_t - \boldsymbol{\theta}_{PS} \right\|^2 - L(1+\beta) \cdot \left( \frac{\tilde{\mu}}{2L(1+\beta)} \left\| \widetilde{\boldsymbol{\theta}}_t - \boldsymbol{\theta}_{PS} \right\|^2 + \frac{2L(1+\beta)}{\tilde{\mu}} \gamma_t^2 \| e_t \|^2 \right)$$

$$= \frac{\tilde{\mu}}{2} \left\| \widetilde{\boldsymbol{\theta}}_t - \boldsymbol{\theta}_{PS} \right\|^2 - \frac{2L^2(1+\beta)^2}{\tilde{\mu}} \gamma_t^2 \| e_t \|^2,$$

where in inequality $(a)$, we have used A2, in inequality (b), we have used A5 and Lemma D.4 of (Perdomo et al., 2020), and in equality $(c)$, we use the definition of $\widetilde{\boldsymbol{\theta}}_t = \boldsymbol{\theta}_t - \gamma_t e_t$.

Substituting above upper bound for $B_1$ and $B_2$ to (41) gives us

$$\mathbb{E}_t \left\| \widetilde{\boldsymbol{\theta}}_{t+1} - \boldsymbol{\theta}_{PS} \right\|^2 \leq \left( 1 - \tilde{\mu}\gamma_{t+1} + 2B^2\gamma_{t+1}^2 + 4(\gamma_t - \gamma_{t+1}) \right) \left\| \widetilde{\boldsymbol{\theta}}_t - \boldsymbol{\theta}_{PS} \right\|^2 + 2(G^2 + d\sigma_{\mathsf{DP}}^2)\gamma_{t+1}^2$$
$$+ \left( 6(\gamma_t - \gamma_{t+1})^2 + \frac{4L^2(1+\beta)^2}{\tilde{\mu}} \gamma_t^2 \gamma_{t+1} \right) \| e_t \|^2. \tag{43}$$

Setting $\gamma_{t+1} \leq \tilde{\mu}/(4B^2)$ ensures that

$$\mathbb{E}_t \left\| \widetilde{\boldsymbol{\theta}}_{t+1} - \boldsymbol{\theta}_{PS} \right\|^2 \leq \left( 1 - (\tilde{\mu}/2)\gamma_{t+1} + 4(\gamma_t - \gamma_{t+1}) \right) \left\| \widetilde{\boldsymbol{\theta}}_t - \boldsymbol{\theta}_{PS} \right\|^2 + 2(G^2 + d\sigma_{\mathsf{DP}}^2)\gamma_{t+1}^2$$
$$+ \left( 6(\gamma_t - \gamma_{t+1})^2 + \frac{4L^2(1+\beta)^2}{\tilde{\mu}} \gamma_t^2 \gamma_{t+1} \right) \| e_t \|^2. \tag{44}$$

If $\gamma_t \equiv \gamma$ is constant step size, then $\gamma_t - \gamma_{t+1} = 0$. For diminishing step size case $\gamma_t = \frac{a_0}{a_1+t}$, if $a_0 \geq \frac{1}{b}$, $b > 0$, we can prove that $\gamma_t - \gamma_{t+1} \leq b\gamma_{t+1}^2$. Therefore, (44) becomes

$$\mathbb{E}_t \left\| \widetilde{\boldsymbol{\theta}}_{t+1} - \boldsymbol{\theta}_{PS} \right\|^2 \leq \left( 1 - (\tilde{\mu}/2)\gamma_{t+1} + 4b\gamma_{t+1}^2 \right) \left\| \widetilde{\boldsymbol{\theta}}_t - \boldsymbol{\theta}_{PS} \right\|^2 + 2(G^2 + d\sigma_{\mathsf{DP}}^2)\gamma_{t+1}^2$$
$$+ \left( 6b^2\gamma_{t+1}^4 \frac{4L^2(1+\beta)^2}{\tilde{\mu}} \gamma_t^2 \gamma_{t+1} \right) \| e_t \|^2.$$

If $\sup_{t \geq 1} \gamma_t \leq \frac{\tilde{\mu}}{16b}$ then we have $1 - (\tilde{\mu}/2)\gamma_{t+1} + 4b\gamma_{t+1}^2 \leq 1 - (\tilde{\mu}/4)\gamma_{t+1}$. Thus,

$$\mathbb{E}_t \left\| \widetilde{\boldsymbol{\theta}}_{t+1} - \boldsymbol{\theta}_{PS} \right\|^2 \leq \left( 1 - (\tilde{\mu}/4)\gamma_{t+1} \right) \left\| \widetilde{\boldsymbol{\theta}}_t - \boldsymbol{\theta}_{PS} \right\|^2 + 2(G^2 + d\sigma_{\mathsf{DP}}^2)\gamma_{t+1}^2$$
$$+ \left( 6b^2\gamma_{t+1}^4 + \frac{4L^2(1+\beta)^2}{\tilde{\mu}} \gamma_t^2 \gamma_{t+1} \right) \| e_t \|^2.$$

Taking full expectation on both sides and applying A8 leads to

$$\mathbb{E} \left\| \widetilde{\boldsymbol{\theta}}_{t+1} - \boldsymbol{\theta}_{PS} \right\|^2 \leq \left( 1 - (\tilde{\mu}/4)\gamma_{t+1} \right) \left\| \widetilde{\boldsymbol{\theta}}_t - \boldsymbol{\theta}_{PS} \right\|^2 + 2(G^2 + d\sigma_{\mathsf{DP}}^2)\gamma_{t+1}^2 + 6b^2M^2\gamma_{t+1}^4$$
$$+ \frac{4L^2M^2(1+\beta)^2}{\tilde{\mu}} \gamma_t^2 \gamma_{t+1}.$$

Suppose that $\sup_{t \geq 1} \gamma_t^2/\gamma_{t+1}^2 \leq 1 + \bar{b}\gamma_{t+1}^2$, then we have

$$\mathbb{E} \left\| \widetilde{\boldsymbol{\theta}}_{t+1} - \boldsymbol{\theta}_{PS} \right\|^2 \leq \left( 1 - (\tilde{\mu}/4)\gamma_{t+1} \right) \left\| \widetilde{\boldsymbol{\theta}}_t - \boldsymbol{\theta}_{PS} \right\|^2 + 2(G^2 + d\sigma_{\mathsf{DP}}^2)\gamma_{t+1}^2 + \frac{4L^2M^2(1+\beta)^2}{\tilde{\mu}} \gamma_{t+1}^3$$
$$+ 6b^2M^2\gamma_{t+1}^4 + \frac{4L^2M^2\bar{b}(1+\beta)^2}{\tilde{\mu}} \gamma_{t+1}^5.$$

Solving the above recursion leads to

$$
\mathbb{E}\left\|\widetilde{\boldsymbol{\theta}}_{t+1} - \boldsymbol{\theta}_{PS}\right\|^2 \leq \prod_{i=1}^{t+1}(1-(\tilde{\mu}/4)\gamma_i)\left\|\widetilde{\boldsymbol{\theta}}_0 - \boldsymbol{\theta}_{PS}\right\|^2 + 2(G^2 + d\sigma_{\mathsf{DP}}^2)\sum_{i=1}^{t+1}\gamma_i^2\prod_{j=i+1}^{t+1}(1-\frac{\tilde{\mu}}{4}\gamma_i)
$$

$$
+ \frac{4L^2M^2(1+\beta)^2}{\tilde{\mu}}\sum_{i=1}^{t+1}\gamma_i^3\prod_{j=i+1}^{t+1}(1-\frac{\tilde{\mu}}{4}\gamma_i) + 6b^2M^2\sum_{i=1}^{t+1}\gamma_i^4\prod_{j=i+1}^{t+1}(1-\frac{\tilde{\mu}}{4}\gamma_i)
$$

$$
+ \frac{4L^2M^2\bar{b}(1+\beta)^2}{\tilde{\mu}}\sum_{i=1}^{t+1}\gamma_i^5\prod_{j=i+1}^{t+1}(1-\frac{\tilde{\mu}}{4}\gamma_i)
$$

$$
\leq \prod_{i=1}^{t+1}(1-(\tilde{\mu}/4)\gamma_i)\left\|\widetilde{\boldsymbol{\theta}}_0 - \boldsymbol{\theta}_{PS}\right\|^2 + \frac{8(G^2 + d\sigma_{\mathsf{DP}}^2)}{\tilde{\mu}}\gamma_{t+1} + \frac{16L^2M^2(1+\beta)^2}{\tilde{\mu}^2}\gamma_{t+1}^2
$$

$$
+ \frac{24b^2M^2}{\tilde{\mu}}\gamma_{t+1}^3 + \frac{16L^2M^2\bar{b}(1+\beta)^2}{\tilde{\mu}^2}\gamma_{t+1}^4, \tag{45}
$$

where the last inequality follows from Lemma 1. Thus, we can now conclude the proof.

## G. Proof of Theorem 8

**Theorem 8 (formal).** *Under A2, 4, 6, 7, 8. Let $\gamma$ denotes the constant step size. If it holds that $\gamma \leq 1/(2L(1+\sigma_1^2))$, then the iterates generated by* **DiceSGD** *admits the following bound for any $T \geq 1$,*

$$
\min_{t=0,...,T-1}\mathbb{E}\left[\|\nabla f(\boldsymbol{\theta}_t;\boldsymbol{\theta}_t)\|^2\right] \leq \frac{4\Delta_0}{T\gamma} + \mathsf{b}\beta + 2L\gamma\left[\sigma_{\mathsf{DP}}^2 + \sigma_0^2\right] + 2L^2M^2\gamma^2, \tag{46}
$$

*where $\Delta_0 := \mathbb{E}\left[f(\boldsymbol{\theta}_0 - \gamma e_0;\boldsymbol{\theta}_0) - \ell^\star\right]$ is an upper bound for the initial error and the constant $\mathsf{b}$ is defined as*

$$
\mathsf{b} := 4\ell_{\max}\left(C_1 + C_2 + \sqrt{d}\sigma_{\mathsf{DP}}\right).
$$

*In particular, for sufficiently large $T$, setting $\gamma = 1/\sqrt{T}$ yields the bound in* (21) *of the main paper.*

*Proof.* [2]Consider the **DiceSGD** algorithm with constant step size $\gamma_{t+1} = \gamma$. Similar to §F, we define $\widetilde{\boldsymbol{\theta}}_t = \boldsymbol{\theta}_t - \gamma e_t$ and notice the following recursion:
$$
\widetilde{\boldsymbol{\theta}}_{t+1} = \widetilde{\boldsymbol{\theta}}_t - \gamma\left(\zeta_{t+1} + \nabla\ell(\boldsymbol{\theta}_t; Z_{t+1})\right)
$$
To facilitate the derivations, we define the shorthand notation: $\nabla f_t := \nabla f(\boldsymbol{\theta}_t;\boldsymbol{\theta}_t)$. For any fixed sample $z \in \mathsf{Z}$, applying A2 leads to the following upper bound:

$$
\ell(\widetilde{\boldsymbol{\theta}}_{t+1}, z) \leq \ell(\widetilde{\boldsymbol{\theta}}_t, z) + \langle\nabla\ell(\widetilde{\boldsymbol{\theta}}_t, z)\,|\,\widetilde{\boldsymbol{\theta}}_{t+1} - \widetilde{\boldsymbol{\theta}}_t\rangle + \frac{L}{2}\left\|\widetilde{\boldsymbol{\theta}}_{t+1} - \widetilde{\boldsymbol{\theta}}_t\right\|^2
$$

Taking integration on the fixed $z$ with weights given by the p.d.f. of $\mathcal{D}(\boldsymbol{\theta}_t)$ on the both sides of above inequality yields

$$
f(\widetilde{\boldsymbol{\theta}}_{t+1}, \boldsymbol{\theta}_t) \leq f(\widetilde{\boldsymbol{\theta}}_t, \boldsymbol{\theta}_t) + \langle\nabla f(\widetilde{\boldsymbol{\theta}}_t, \boldsymbol{\theta}_t)\,|\,\widetilde{\boldsymbol{\theta}}_{t+1} - \widetilde{\boldsymbol{\theta}}_t\rangle + \frac{L}{2}\left\|\widetilde{\boldsymbol{\theta}}_{t+1} - \widetilde{\boldsymbol{\theta}}_t\right\|^2
$$

$$
= f(\widetilde{\boldsymbol{\theta}}_t, \boldsymbol{\theta}_t) - \gamma\langle\nabla f(\widetilde{\boldsymbol{\theta}}_t, \boldsymbol{\theta}_t)\,|\,\zeta_{t+1} + \nabla\ell(\boldsymbol{\theta}_t; Z_{t+1})\rangle + \frac{L\gamma^2}{2}\left\|\zeta_{t+1} + \nabla\ell(\boldsymbol{\theta}_t; Z_{t+1})\right\|^2
$$

Taking the conditional expectation $\mathbb{E}_t[\cdot]$ yields

$$
\mathbb{E}_t[f(\widetilde{\boldsymbol{\theta}}_{t+1}, \boldsymbol{\theta}_t)] \leq f(\widetilde{\boldsymbol{\theta}}_t, \boldsymbol{\theta}_t) - \gamma\langle\nabla f(\widetilde{\boldsymbol{\theta}}_t, \boldsymbol{\theta}_t)\,|\,\nabla f(\boldsymbol{\theta}_t;\boldsymbol{\theta}_t)\rangle + \frac{L\gamma^2}{2}\mathbb{E}_t\left[\left\|\zeta_{t+1} + \nabla\ell(\boldsymbol{\theta}_t; Z_{t+1})\right\|^2\right] \tag{47}
$$

---

[2]We notice that our proof differs from that of (Zhang et al., 2024, Theorem 3.6) and is simpler than the latter due to the addition of A8.

Notice that

$$\mathbb{E}_t \left[ \| \zeta_{t+1} + \nabla \ell(\boldsymbol{\theta}_t; Z_{t+1}) \|^2 \right] \leq \sigma_{\mathsf{DP}}^2 + \sigma_0^2 + (1 + \sigma_1^2) \| \nabla f(\boldsymbol{\theta}_t; \boldsymbol{\theta}_t) \|^2$$

and under A8, it holds that

$$
\begin{aligned}
-\mathbb{E}\langle \nabla f(\widetilde{\boldsymbol{\theta}}_t, \boldsymbol{\theta}_t) \,|\, \nabla f(\boldsymbol{\theta}_t; \boldsymbol{\theta}_t) \rangle &= -\mathbb{E}\|\nabla f_t\|^2 + \mathbb{E}\langle \nabla f(\widetilde{\boldsymbol{\theta}}_t, \boldsymbol{\theta}_t) - \nabla f(\boldsymbol{\theta}_t; \boldsymbol{\theta}_t) \,|\, \nabla f(\boldsymbol{\theta}_t; \boldsymbol{\theta}_t) \rangle \\
&\leq -\frac{1}{2}\mathbb{E}\|\nabla f_t\|^2 + \frac{1}{2}\mathbb{E}\|\nabla f(\widetilde{\boldsymbol{\theta}}_t, \boldsymbol{\theta}_t) - \nabla f(\boldsymbol{\theta}_t; \boldsymbol{\theta}_t)\|^2 \\
&\stackrel{(a)}{\leq} -\frac{1}{2}\mathbb{E}\|\nabla f_t\|^2 + \frac{L^2}{2}\mathbb{E}\|\widetilde{\boldsymbol{\theta}}_t - \boldsymbol{\theta}_t\|^2 \\
&\leq -\frac{1}{2}\mathbb{E}\|\nabla f_t\|^2 + \frac{L^2}{2}\mathbb{E}\|\gamma e_t\|^2 \leq -\frac{1}{2}\mathbb{E}\|\nabla f_t\|^2 + \frac{L^2 M^2 \gamma^2}{2}
\end{aligned}
$$

where $(a)$ is due to Lemma 2. Substituting back into (47) and taking full expectation lead to

$$
\begin{aligned}
\mathbb{E}[f(\widetilde{\boldsymbol{\theta}}_{t+1}, \boldsymbol{\theta}_t)] &\leq \mathbb{E}[f(\widetilde{\boldsymbol{\theta}}_t, \boldsymbol{\theta}_t)] - \frac{\gamma}{2}\left(1 - L\gamma(1 + \sigma_1^2)\right)\mathbb{E}[\|\nabla f_t\|^2] + \frac{L\gamma^2}{2}\left[\sigma_{\mathsf{DP}}^2 + \sigma_0^2\right] + \frac{L^2 M^2 \gamma^3}{2} \\
&\leq \mathbb{E}[f(\widetilde{\boldsymbol{\theta}}_t, \boldsymbol{\theta}_t)] - \frac{\gamma}{4}\mathbb{E}[\|\nabla f_t\|^2] + \frac{L\gamma^2}{2}\left[\sigma_{\mathsf{DP}}^2 + \sigma_0^2\right] + \frac{L^2 M^2 \gamma^3}{2}
\end{aligned}
$$

where the last inequality is due to $\gamma \leq 1/(2L(1 + \sigma_1^2))$. Recall that we have bounded a similar term in (36):

$$\mathbb{E}\left[ f(\widetilde{\boldsymbol{\theta}}_{t+1}; \boldsymbol{\theta}_t) - f(\widetilde{\boldsymbol{\theta}}_{t+1}; \boldsymbol{\theta}_{t+1}) \right] \leq \ell_{\max}\beta \|\boldsymbol{\theta}_t - \boldsymbol{\theta}_{t+1}\| \leq \gamma \ell_{\max}\beta \left( (C_1 + C_2) + \sqrt{d}\sigma_{\mathsf{DP}} \right),$$

under A6 & 7. This yields

$$\frac{\gamma}{4}\mathbb{E}[\|\nabla f_t\|^2] \leq \mathbb{E}[f(\widetilde{\boldsymbol{\theta}}_t, \boldsymbol{\theta}_t) - f(\widetilde{\boldsymbol{\theta}}_{t+1}, \boldsymbol{\theta}_{t+1})] + \gamma \ell_{\max}\beta \left( C_1 + C_2 + \sqrt{d}\sigma_{\mathsf{DP}} \right) + \frac{L\gamma^2}{2}\left[\sigma_{\mathsf{DP}}^2 + \sigma_0^2\right] + \frac{L^2 M^2 \gamma^3}{2}$$

Summing up the inequality from $t = 0$ to $t = T - 1$ and dividing by $T\gamma/4$ yields

$$\frac{1}{T}\sum_{t=0}^{T-1}\mathbb{E}[\|\nabla f_t\|^2] \leq \frac{4}{T\gamma}\left[ f(\widetilde{\boldsymbol{\theta}}_0, \boldsymbol{\theta}_0) - \ell^\star \right] + 4\ell_{\max}\beta\left( C_1 + C_2 + \sqrt{d}\sigma_{\mathsf{DP}} \right) + 2L\gamma\left[\sigma_{\mathsf{DP}}^2 + \sigma_0^2\right] + 2L^2 M^2 \gamma^2$$

For sufficiently large $T$, setting $\gamma = 1/\sqrt{T}$ yields the bound in the desired theorem. $\qquad\square$

# H. Details of Numerical Experiments

This section provides additional details for the numerical experiments in §5.

## H.1. Validating A8 Empirically

Fig. 3 plots the average of $\|e_t\|^2$ for **DiceSGD** as a function of $t$. Observe that $\|e_t\|^2$ is always bounded for both of our numerical examples, thus validating our A8 empirically.

## H.2. Additional Details of Numerical Experiments

We provide additional details for the numerical experiments on the logistics regression problem.

**Detailed Setup.** We develop a strategic classification example adapted from (Perdomo et al., 2020, Sec. 5). Consider the learner as a bank operator who aims to minimize the logistic loss given by

$$\ell(\boldsymbol{\theta}; z) = \alpha(z)\left(\log(1 + \exp(\langle x \,|\, \boldsymbol{\theta} \rangle)) - y\langle x \,|\, \boldsymbol{\theta} \rangle\right) + \eta/2\, \|\boldsymbol{\theta}\|^2, \tag{48}$$

where $\eta > 0$ is a regularization parameter, $z \equiv (x, y) \in \mathbb{R}^d \times \{0, 1\}$ is the training sample, $\alpha(z) = y + 1$ is a label weight, $y = 0$ ($y = 1$) denotes a customer without (with) history of defaults.

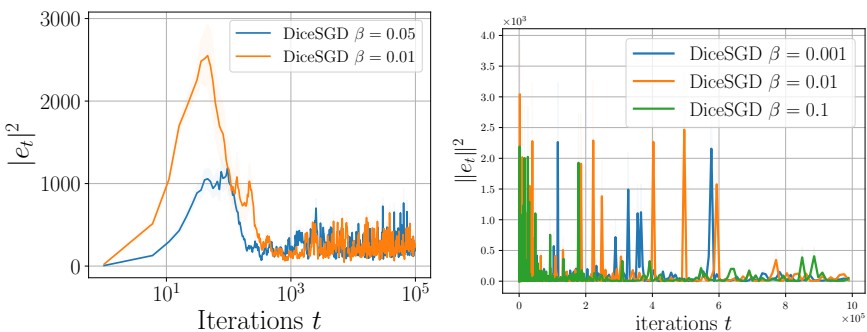

*Figure 3.* Behavior of $e_t$ with **DiceSGD** for quadratic minimization (*left*) and logistic regression (*right*).

Training samples are contributed by $m$ users (e.g., customers served by the bank) that form a base distribution $\mathcal{D}_0 = \mathrm{Unif}(\{(\bar{x}_i, \bar{y}_i)\}_{i=1}^m)$. Upon knowing about the classifier model $\boldsymbol{\theta}$, the customer queried by the bank/learner may shift his/her profile strategically. A possible model is:

$$Z \sim \mathcal{D}(\boldsymbol{\theta}) \Leftrightarrow Z = (x_i, y_i) \text{ with } i \sim \mathrm{Unif}([m]), \ y_i = \bar{y}_i, \ x_i = \arg\max_x U(x; \bar{x}_i, \boldsymbol{\theta}), \tag{49}$$

For example, $U(x; \bar{x}_i, \boldsymbol{\theta}) = \frac{-y_i \langle \boldsymbol{\theta} \mid x \rangle}{\max\{\epsilon_{\mathsf{U}}, \|\boldsymbol{\theta}\|\}} - \frac{1}{2\beta} \|x - \bar{x}_i\|^2$ with $\epsilon_{\mathsf{U}}, \beta > 0$ models strategical users with history of defaults who are trying to evade detection.

**Additional Experiments.** Fig. 4 (Left) plots the performative risk $\mathbb{E}_{z \sim \mathcal{D}(\boldsymbol{\theta}_t)}[\ell(\boldsymbol{\theta}_t; z)]$ as a function of the number of iterations $t$. We observe that **DiceSGD** can achieve lower performative risk at a faster rate compared to **PCSGD**. Furthermore, increasing sensitivity $\beta \uparrow$ leads to lower train true positive rate, which is compatible with our Theorems 3 and 7.

In the training set, records with positive labels ($y = 1$) account for 6.624%, while negative samples ($y = 0$) comprise 93.37%. To evaluate model performance, we have presented the test true negative/positive rate in Fig. 2 (second & third). The training true negative/positive rate is shown in Fig. 4 *(Middle & Right)*. For self-completeness, we recall the definition positive (negative) label accuracy as follows:

$$\text{True Positive (Negative) rate} = \frac{\mathrm{TP}_{\text{pos/neg}}}{\mathrm{TP}_{\text{pos/neg}} + \mathrm{FN}_{\text{pos/neg}}}$$

where $\mathrm{TP}_{\text{pos/neg}}$ denotes the number of samples with positive (negative) labels correctly classified as positive (negative), whereas $\mathrm{FN}_{\text{pos/neg}}$ represents the number of samples with positive (negative) labels incorrectly classified as negative (positive).

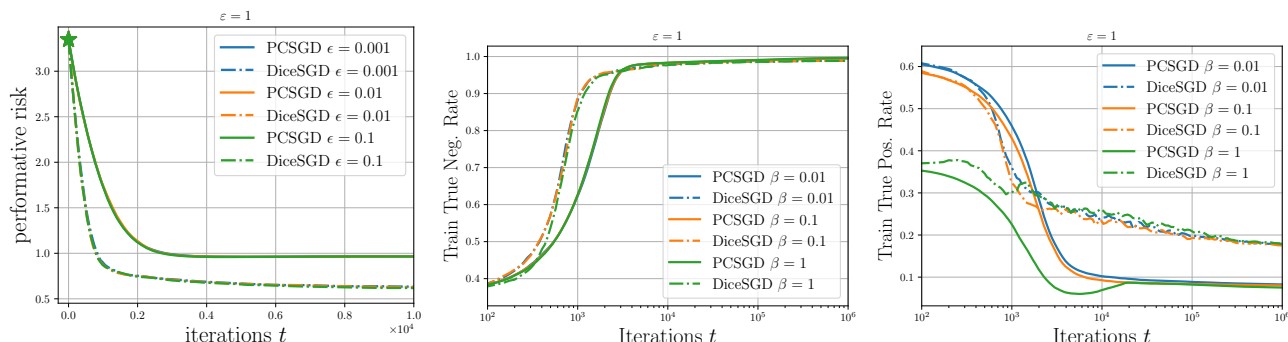

*Figure 4.* **Logistic Regression** *(Left)*: Performative Risk $V(\boldsymbol{\theta})$. *(Middle) & (Right)*: Train true neg./pos. rate.

## H.3. Additional Numerical Experiments on Non-convex Loss

In this subsection, we present additional simulations on non-convex loss based on synthetic data to support Theorems 5 & 8.

**Synthetic Data with Linear Model.** We consider the following binary classification problem using a linear model. We adopt the following sigmoid loss function:

$$\ell(\boldsymbol{\theta}; z) := \left(1 + \exp(c \cdot y x^\top \boldsymbol{\theta})\right)^{-1} + \frac{\eta}{2} \|\boldsymbol{\theta}\|^2, \tag{50}$$

which is smooth but **non-convex** for small regularization parameter $\eta > 0$. The data distribution $\mathcal{D}^o \equiv \{(x_i, y_i)\}_{i=1}^m$ is generated by $\mathrm{Unif}[-1, 1]$, with $m = 10^3$, and the label $\bar{y}_i = \mathrm{sgn}(\langle x_i \,|\, \boldsymbol{\theta}^o \rangle) \in \{\pm 1\}$. The shift dynamics are given by $\{(x_i - \beta \boldsymbol{\theta}; y_i)\}_{i=1}^m$. We randomly flip 10% of the labels to obtain the final labels $y_i$. In detail, we set $d = 10$, $c = 0.1$, and $\beta \in \{0.1, 0.5\}$. The batch size is $b = 1$, and the stepsize is constant, $\gamma = 1/\sqrt{T}$, with $T = 10^6$.

We implement the **PCSGD**, **DiceSGD**, and **Clip21**[3] algorithms (in the $n = 1$ case) (Khirirat et al., 2023). From Fig. (5), we observe that the SPS-stationarity measures in 2 for the algorithms initially decrease and eventually saturate at a certain level, which is consistent with the predictions of Theorems 5 & 8.

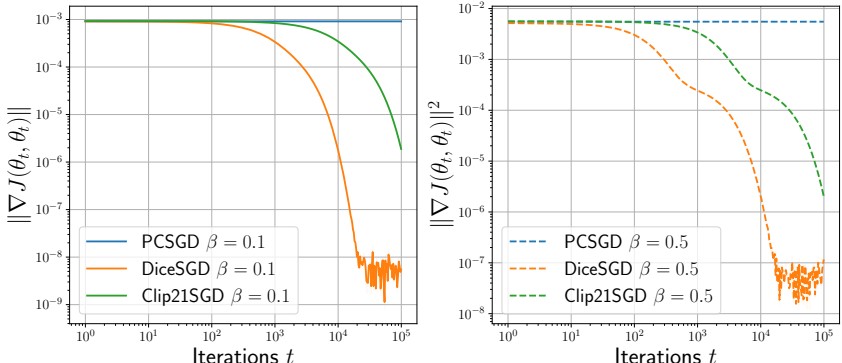

*Figure 5.* **Synthetic Data with Sigmoid Loss.** SPS measure $\|\nabla J(\boldsymbol{\theta}_t; \boldsymbol{\theta}_t)\|^2$ of **PCSGD**, **DiceSGD**, and **Clip21** algorithm with greedy deployment schemes versus iteration number $t$.

---

[3]We set the parameters as following: injected noise variance $\sigma = 1$, two clipping thresholds $\nu = 1$, and $\tau = 6\nu$.

