# OpenReview forum: "Clipped SGD Algorithms for Performative Prediction: Tight Bounds for Stochastic Bias and Remedies"
_ICML.cc/2025/Conference — ICML 2025 poster_

### Official Review · Reviewer_KRfQ · 2025-03-03

**Overall Recommendation:** 3

**Summary:**

The paper studies the problem of Performative Prediction: that is when the data distribution also depends on the model weights. This problem also includes the case of differentially private algorithms. The paper studies clipped SGD for finding a stable solution.  Under various assumptions, the paper provides upper bounds and lower bounds for the distance to the stable solution in the strongly convex case. The algorithm extends to the nonconvex case and differential privacy. The paper further studies the DiceSGD algorithm with two clipping operators. Under the assumption that the error feedback is bounded in expectation, the authors show that the bias from the clipped algorithm goes to 0.

## Update after rebuttal:
The authors have addressed my question. I maintain my positive assessment of the paper.

**Claims And Evidence:**

The theoretical claims appear to be correct. The experimental results support these claims.

**Essential References Not Discussed:**

None that I'm aware of.

**Experimental Designs Or Analyses:**

For the purpose of validating the theoretical claims, I think experimental setup in the paper is sufficient, although it is unclear the reason for the choice of the data distribution.

**Methods And Evaluation Criteria:**

The evaluation is based on simple experiments and small datasets. For the purpose of validating the claims, they are fine. To show actually applications of the algorithm, the paper needs to show on real datasets with clear justifications/considerations for how the decision-dependent distributions are modeled.

**Other Comments Or Suggestions:**

- In the proof of Lemma 3, the authors several times referred to Eq. 7 which didn't match the intended purpose.

**Other Strengths And Weaknesses:**

- Personally I find the presentation hard to follow and I would prefer that the results and theorem statements are presented in a more concise way.

**Questions For Authors:**

- Do you have any guarantees for the non-strongly convex case?
- Is the dependence on the dimension necessary?

**Relation To Broader Scientific Literature:**

The paper studies a problem of recent interest and the paper offers concrete theoretical guarantees for clipped SGD in this setting.

**Theoretical Claims:**

- I checked Theorem 3 and 4, both are correct, except some typos. In the main paper, Theorem 3 made reference to $\sigma_{DP}$ before it was introduced later. In the proof in the appendix, $\zeta_t$ was also used (without proper introduction), and disappear in Line 647-652.
- Validation of Assumption 8: The bound for $|e_t|$ is quite large in Figure 3, the question is how meaningful the bound $M$ is if it is so large.

---

> ### Author Rebuttal · Authors · 2025-03-31
>
> > To demonstrate practical applications of the algorithm, the paper needs to include real datasets with clear justifications/considerations for how the decision-dependent distributions are modeled.
>
> Please refer to the response to Reviewer svSm.
>
> > The appearance of $\zeta_{t}$ and $\sigma_{DP}^2$ in Theorem 3.
>
> Thank you very much for the careful review and we apologize for these careless typos.
>
> Indeed, Theorem 3 should be presented without $\sigma_{DP}^2$, while we remark that the theorem in L579-597 presents a general version of the result. As for the noise $\zeta_{t+1}$, it should appear in the inner product terms in L647-652. That said, it does not affect the next step in (29) as the noise is zero-mean.
>
> > Validation of Assumption 8: The bound for $|e_{t}|$ is quite large in Figure 3. The concern is how meaningful the bound is if it is so large.
>
> The bound is meaningful as we observe in Theorem 7 that the relevant term to it is in the order of $O( M^2 \gamma_t^2 )$ which can be dominated by the $O( \gamma_t )$ term as $\gamma_t$ is diminishing such as when $\gamma_t = O(1/t)$. Particularly, as seen in Fig. 3, the empirical upper bound for $M^2$ is around $10^3$, and its effect on $\| \tilde{\theta}_t \|^2$ becomes negligible ($\approx 10^{-6}$) with $\gamma_t = 50/(5000+t)$, $t \geq 10^6$.
>
> > Experiments: The reason for the choice of the data distribution is unclear.
>
> These choices follow from the literature in performative prediction such as [Perdomo et al, 2020]. For both cases in our experiments, the distribution shifts are designed to model strategic behavior where the data owners (users) maximize a quadratic utility function adapting to the new $\theta$, e.g., for logistics regression for defaults detection, we have $(x^\star,y) \sim {\cal D}(\theta)$ given by:
>
> $x^\star = \arg\max_{x} U(x;\bar{x},\theta) =\frac{- y <{\theta},{x}>}{\max\\{ \epsilon_{\sf U}, \| \theta \| \\}}  - \frac{1}{2\beta} || x-\bar{x} ||^2$
>
> where $(\bar{x},y)$ is drawn from the base distribution. In doing so, users with a default history shift their profile towards having a lower chance to be detected as fraud ($y=1$).
>
> > Typos
>
> Indeed, in the proof of Lemma 3, it should point to (2) instead of (7). We will revise it.
>
> > Do you have any guarantees for the non-strongly convex case?
>
> To our best knowledge, there is no known convergence guarantee for SGD with distribution shift with non-strongly convex loss function, even without clipping. The main challenge is that in the latter case, SGD (or even GD) may not converge to a unique solution even when the distribution shift is fixed.
>
> Having said that, the non-strongly convex case can be partially covered by our Theorem 5 & 8 for the non-convex case, provided that $\{ \sup_{z \in Z} |\ell(\theta_t ; z)| \}_{t \geq 0}$ is bounded. That said, we admit that the obtained bound is weaker than the strongly convex case.
>
>
> > Is the dependence on the dimension necessary?
>
> The dependence on dimensionality arises from the setting that $\zeta_{t+1} \sim {\cal N}(0, \sigma_{DP}^2 {\bf I})$ in (14) which is the same as in (Abadi et al., 2016) for satisfaction of the DP guarantee. This leads to $E[ || \zeta_{t+1} ||^2 ] = d \sigma_{DP}^2$. The same $d$-dependence would appear in Sec 3.3 of (Koloskova et al., 2023) if a similar design for the DP noise is adopted. Note that their paper took $\zeta_{t+1} \sim {\cal N}(0, (\sigma_{DP}^2/d) {\bf I})$.

---

### Official Review · Reviewer_5Ee1 · 2025-03-13

**Overall Recommendation:** 3

**Summary:**

The paper claims error bounds for the estimate obtained using projected clipped SGD (PCSGD) and DiceSGD algorithm in the problem of performative prediction, where an artificial noise can be added to preserve data privacy. While PCSGD is known for its stability, the output of the algorithm exhibits bias from the performative stable solution in this setting, as its lower and upper bound is proved in the paper. The expected bounds are also proven for DiceSGD, where its result is asymptotically unbiased.

## Update after rebuttal

My questions about the paper have been clarified, and I intend to maintain my original recommendation score.

**Claims And Evidence:**

The expected error bounds of PCSGD and DiceSGD claimed in the proposed theorems are supported with proofs and numerical experiments, along with the effect of differential data privacy.

**Essential References Not Discussed:**

No special reference I would like to discuss.

**Experimental Designs Or Analyses:**

The designs for the numerical experiments sound valid. However, it seems there is no setting that the loss function is nonconvex, so the experiments reported does not validate the claim made regarding nonconvex loss (Theorem 5).

**Methods And Evaluation Criteria:**

The evaluated bias reported in the experiment result goes along with the claims made in the theory section about the estimation bias and differential privacy.

**Other Comments Or Suggestions:**

No additional comments.

**Other Strengths And Weaknesses:**

In the both PCSGD and DiceSGD, the randonmized Gaussian noise $\zeta \sim N(0, \sigma_{DP}^2I)$ is employed for differential privacy guarantees, but the proposed $\sigma_{DP}^2$ for guraunteeing $(\epsilon,\delta)$-differential privacy also depends on the number of iterations $T$. It may be difficult to choose appropriate magnitude of the noise if the number of appropriate SGD iterations is not available at the first hand.

**Questions For Authors:**

1. Regarding to the result of the quadratic minimization (Figure 1), what is the randomized mechanism for differential privacy used in here? How exactly does the value of privacy budget $\epsilon$ effect the training?

2. Is there a particular reason that the clipped SGD is selected for performative prediction? If for stability, for example, other methods can be considered such as implicit SGD [1].

[1] Lee, Y., Lee, S., & Won, J. H. (2022, June). Statistical inference with implicit sgd: proximal robbins-monro vs. polyak-ruppert. In International Conference on Machine Learning (pp. 12423-12454). PMLR.

**Relation To Broader Scientific Literature:**

Data privacy issues are becoming an important issue in the digital age nowadays. For a more performant model, the size of the training data and the model itself are increasing, which also raises the risk of exposing individual information within the training data. It will become increasingly necessary to explore training methods that protect data privacy at a minimal cost to model performance.

**Theoretical Claims:**

Theoretical claims seem sound.

---

> ### Author Rebuttal · Authors · 2025-03-31
>
> > Experiments: There is no setting where the loss function is nonconvex, so the reported experiments do not validate the claim regarding nonconvex loss (Theorem 5).
>
>
> We conducted an additional binary classification experiment in **[https://ibb.co/Jjx0XMRg ]**, where we simulated PCSGD, DiceSGD, Clip21 [Khirirat et al., 2024] under different $\beta$-sensitivity. We adopted the sigmoid loss function:
>
> $$
> \ell(\theta; z) = (1 + \exp(c \cdot y x^\top \theta))^{-1} + \frac{\eta}{2} \|| \theta \||^2
> $$
>
> which is smooth but **non-convex**. The data distribution ${\cal D}^o$ is generated by $Unif[-1, 1]$, and the label $y_i \in \\{ \pm 1 \\}$. The shift dynamics are given by $\\{(x_i - \beta \theta; y_i)\\}_{i=1}^{1e3}$. We observed that the SPS stationarity measures in (3) of the algorithms initially decrease and eventually saturate at a certain level, which is consistent with the predictions made by Theorems 5 and 8.
>
>
>
> > In both PCSGD and DiceSGD, the noise $\zeta \sim \mathcal{N}(0, \sigma^2_{DP}I)$ is employed for DP, but $\sigma_{DP}^2$ depends on $T$. It may be difficult to choose an appropriate $\sigma_{DP}$ if the number of required SGD iterations is not known in advance.
>
> This is a good observation. However, we believe that obtaining a DP bound independent of $T$ would be difficult under the current setting. The reason is that the potential attacker have access to the training history of $T$ iterations. The larger $T$ is, intuitively there will be more privacy leaks. We observe similar conclusions in [Abadi et al., 2016], [Zhang et al., 2024].
>
> > In Figure 1, what randomized mechanism for differential privacy is used? How exactly does the value of the privacy budget affect training?
>
> We are using the Gaussian mechanism, see (14), for DP. In our numerical exp, the privacy budget affects the DP noise variance as studied in Corollary 1.
>
> > Is there a particular reason why Clipped-SGD is selected for performative prediction? If it is for stability, for example, other methods such as implicit SGD [1] could also be considered.
>
> The clipped SGD algorithm is selected in our context due to its popularity and wide range of applications, e.g., DP enabled training, robust optimization, etc. This motivated us to analyze its convergence behavior in the context of performative prediction, where the distribution shifts may occur naturally in several key applications of clipped SGD.
>
> Similar to clipped SGD which introduced non-smooth behavior to the stochastic process, we believe that analyzing algorithms such as Implicit SGD in the performative prediction setting is an interesting direction. Thank you for the suggestion.

---

> > ### Comment · Reviewer_5Ee1 · 2025-04-07
> >
> > Thank you for your thorough and detailed response. My questions about the paper have been clarified, and I intend to maintain my original recommendation score.

---

### Official Review · Reviewer_ETQW · 2025-03-19

**Overall Recommendation:** 3

**Summary:**

This paper examines the convergence behavior of clipped stochastic gradient descent algorithms in the performative prediction setting, where the subsampling distribution depends on the previous iterate. The theoretical analysis addresses both strongly convex and non-convex objective functions.

**Claims And Evidence:**

The claim is clearly stated and well-supported by both theoretical proofs and empirical results.

**Essential References Not Discussed:**

Please refer to my response to Question 2.

**Experimental Designs Or Analyses:**

The experimental design seems reasonable to me.

**Methods And Evaluation Criteria:**

Yes

**Other Comments Or Suggestions:**

* Lines 301–302: The claim that the algorithm achieves $(\varepsilon, \delta)$-Rényi Differential Privacy is not standard. Typically, Rényi DP (RDP) is parameterized solely by $(\alpha, \varepsilon)$, where $\alpha > 1$ is the Rényi order. The inclusion of a $\delta$ parameter suggests a composition or conversion to approximate DP (i.e., $(\varepsilon, \delta)$-DP), which should be clarified. Please revise the statement to reflect the standard formulation of RDP or explain the intended interpretation.


* Algorithm 1, Line 1: Is $e_t$ a scalar or a vector? The notation is ambiguous and could benefit from clarification. If $e_t$ represents a step size, noise scale, or another parameter, please specify its role and dimensionality explicitly to avoid confusion.

**Other Strengths And Weaknesses:**

Strengths:
* The paper provides a rigorous analysis of clipping bias and convergence in the performative prediction setting, addressing both convex and non-convex objectives.

Weaknesses:
* The analytical techniques employed in this paper are largely grounded in existing literature. For example, the privacy accounting builds on the moment accountant framework proposed by Abadi et al. (2016), while the convergence analysis draws from methodologies developed by Li et al. (2022), Perdomo et al. (2020), and Zhang et al. (2024).

* The treatment of distribution shift is based on a worst-case bound. It would be valuable to further investigate the interaction between distribution shift and differential privacy, as this could uncover additional insights into the practical implications of the proposed algorithms.

**Questions For Authors:**

* The paper considers two cases: strongly convex and non-convex objectives. Would similar theoretical guarantees or insights extend to the convex (but not strongly convex) case? Clarifying whether the techniques or results generalize to this intermediate setting would strengthen the contribution.

* I was unable to locate a dedicated related work section. Are there key prior works the authors could discuss to better contextualize their contributions? A comparison to existing results in differentially private optimization—particularly those involving clipping techniques—would be valuable

* Could the author further clarify the novelty of this work? The contribution remains somewhat unclear, particularly regarding the core message the paper intends to convey. If the primary goal is to analyze clipped (stochastic) gradient descent, then what is the specific technical contribution? It appears that certain assumptions—especially Assumptions A.5 and A.8—simplify the analysis and allow the application of existing techniques for studying clipped gradient descent.

* Continuing from the previous point, it remains unclear whether clipped-SGD-based performative prediction is significantly different from standard analyses of clipped SGD under Assumption A.5. The authors are encouraged to clarify this distinction and elaborate on whether the performative aspect introduces fundamentally new challenges or insights beyond existing clipped SGD frameworks.

**Relation To Broader Scientific Literature:**

(DP) Optimization

**Theoretical Claims:**

Yes

---

> ### Author Rebuttal · Authors · 2025-03-31
>
> > Difference in analysis vs existing works.
>
> Indeed, some of the techniques we applied are standard and grounded in existing works, yet we emphasize that this is the first rigorous study of clipping bias with performative prediction. The lower bounding result in Theorem 4 is new as it emphasizes on the tightness of the upper bounds in Theorem 3. It shows that bias analysis in Theorem 3 is unlikely to be improved with different analysis. We elaborate more on this point in our response to your last comment.
>
> > Interaction between distribution shift and differential privacy beyond worst case
>
> We agree, especially as the co-evolution of DP clipped SGD algorithm and distribution shift is an exciting venue for future investigation. That said, our results derived from worst case sensitivity offers a faithful approximation of possibly non-linear interaction. Evidences are (i) the form of distribution shift with linear dependence on $\theta$, which achieves the worst case sensitivity bound, is motivated by a best response dynamics with utility maximization [Perdomo et al., 2020], also see response to rev. KRfQ; (ii) Theorems 3, 4 show that our analysis under the sensitivity assumption is tight.
>
> > Statement about Rényi-DP and $e_t$.
>
> We apologize for the careless typos.
>
> - The discussion about Renyi-DP in L302 (right col) refers to $(\epsilon,\delta)$-DP as in Def. 6. In Appendix A.2 [Zhang et al., 2024], the DiceSGD is first shown to satisfy Renyi-DP, and the standard DP property is established subsequently.
> - The term $e_t$ is in $R^d$. We will revise notations throughout the paper.
>
> > Q: Guarantees for the convex case.
>
> We refer to the response to Rev. KRfQ. In short, Theorems 5 & 8 can partially cover the non-strongly-convex case, yet the analysis is still an open problem for performative prediction using SGD in general, even without clipping.
>
> > Q: Comparison with existing results involving clipping techniques in DP optimization.
>
> Our work is the first to address the convergence of clipped SGD in performative prediction. Below is a brief comparison to related works:
>
> Vs. [Mendler-Dunner et al., 2020] & others on performative prediction -- We conduct analysis for SGD with distribution shifts to clipped SGD which entails a non-continuous mean-field. Such analysis is the first of its kind to our knowledge.
>
> Vs. [Koloskova et al., 2023] & others on clipped SGD -- We study the effects of distribution shift on clipped SGD dynamics with a non-gradient mean field prior to clipping. One new finding is that the clipping bias crucially depend on the strength of distribution shift.
>
> Subject to space limitation, we will include a detailed comparison in the revised version.
>
> > Q: Clarify (i) novelty and main message, (ii) distinction between Clipped-SGD-based performative prediction and standard analyses of Clipped-SGD, and elaborate on the effect of performativity.
>
> Our main contribution and novelty lie in conducting the first rigorous study of *clipped SGD algorithms in the presence of distribution shift (i.e., performative prediction)*. We show that there is an inevitable bias that can be excaberated by distribution shifts, and show that DiceSGD can remedy the bias by extending the latter analysis. Besides the above takeaways for practitioners, here are some novel technical challenges overcome by this work:
>
> - For the stochastic processes generated by PCSGD or DiceSGD, the clipping operator turns the dynamics into a non-continuous one, and distribution shift introduces non-gradient mean-field behavior, it is not possible to adopt existing analysis to work out the convergence analysis.
> - For PCSGD, a notable challenge is in Lemma 4 that controls the error of clipped stochastic gradient $b_t = clip_{c}( \nabla \ell(\theta;Z) ) - \nabla f( \theta; \theta ))$. Existing work such as [Zhang et al., 2020b] only took a crude bound insufficient for our analysis. Our final result yields a bias of $(max\\{G-c,0\\})^2$ that becomes bias-free with large $c$. As comparison, the bias in [Koloskova et al., 2023] is $O( min\\{ \sigma, \sigma^2/c \\})$ which does not vanish even if $c>G$.
> - For DiceSGD, existing result in [Zhang et al., 2024] only works for constant step and non-convex setting without distribution shifts. We made a major revamp including (38) to handle time varying step, (42) to handle non-gradient pre-clipped mean field, etc.
> - We also respectfully disagree with that using A5 & A8 "simplify the analysis and allow the application of existing techniques". A5 is general assumption used in almost all papers on performative prediction, it does not directly simplify the analysis as the effects of distribution shift on PCSGD recursion is non-explicit. A8 is used in controlling the error in DiceSGD in (42), again its role is not immediately clear from a direct inspection. The non-convex loss case also require a non-obvious design for Lyapunov function as $f( \tilde{\theta}, \theta )$, see Appendix G.

---

### Official Review · Reviewer_svSm · 2025-03-21

**Overall Recommendation:** 3

**Summary:**

In this work the authors study the convergence of clipped stochastic gradient descent (SGD) algorithms with decision-dependent data distribution. They explain the performative prediction problem, which is a more general and challenging problem than standard optimization, and they define the performative stable (PS) and stationary PS (SPS) solutions of this. In order find a PS solution they introduce the Projected Clipped SGD (PCSGD) and provide convergence guarantees of this algorithm for stongly convex and general non convex objectives. They also extend their analysis in the Differential Privacy (DP) setting. Additionally, they adapt the previously introduced DiceSGD algorithm for the performative prediction problem and they also offer convergence guarantees of this algorithm for stongly convex and general non convex objectives. Finally, the authors provide numerical experiments applying their algorithms PCSGD and DiceSGD in quadratic minimization and logistic regression problems.

**Claims And Evidence:**

Most of the claims are clear with supporting proofs, however I think the presentation of the theorems needs some improvement. For example, in Theorem 3, what is $\sigma\_{DP}^2$? I understand what $\sigma\_{DP}^2$ is in the context of Section 3.2, however it has not been introduced earlier.

**Essential References Not Discussed:**

All essential references are discussed.

**Experimental Designs Or Analyses:**

The authors provide numerical experiments applying their algorithms PCSGD and DiceSGD in quadratic minimization and logistic regression problems focusing only on the DP setting. In the logistic regression experiments, they only try PCSGD/DiceSGD with different $\beta$. I would expect to have comparisons with other algorithms as well. It would also be interesting to see a few more real-world experiments, for example in the intro, you mention "A common application scenario is that the training data involve human input that responds strategically to the model"

**Methods And Evaluation Criteria:**

The authors propose the PCSGD and DiceSGD to solve the performance prediction problem and they provide guarantees that their algorithms converge in a PS solution up to a neighborhood. My main concern is the number of assumptions they make to guarantee convergence.

For example, Theorem 3 assumes strong convexity, $\beta$-sensitivity, smoothness *and* bounded gradients (Lipschitzness). In contrast, [Perdomo et al, 2020, Thm 3.5] assume strong convexity, $\beta$-sensitivity, and smoothness. Assuming both smoothness and bounded gradients is too strong.

Another example, Theorem 4 assumes smoothness, some kind of bounded variance, bounded gradients (Lipschitzness) and boundedness of $\ell$ on top of the standard $\beta$-sensitivity. Again if we compare with [Koloskova et al, 2023, Thm: 3.3] (essentially $\beta=0$) they only assume smoothness and bounded variance.

**Other Comments Or Suggestions:**

None.

**Other Strengths And Weaknesses:**

Strengths:
1. Propose novel methods, PCSGD and DiceSGD, to solve the performance prediction problem.
2. Provide convergence guarantees for these methods.
3. Validate their theory via numerical experiments.

Weaknesses:
1. What is $\sigma_{DP}^2$ in Theorem 3? (see Claims And Evidence/Theoretical Claims)
2. The authors make many more assumptions for their convergence results compared to the literature (see Methods And Evaluation Criteria)
3. There could be a few more experiments (See Experimental Designs Or Analyses)

**Questions For Authors:**

See weaknesses.

**Relation To Broader Scientific Literature:**

The authors propose two new methods, PCSGD and DiceSGD, for solving the performative prediction problem. This problem was introduced and solved in [Perdomo et al, 2020]. The clipped SGD has been studied extensively in the literature and DiceSGD was introduced in [Zhang et al, 2024]. A recurring issue is that the convergence guarantees that the authors provide, make a lot more assumptions than either of the previous works.

**Theoretical Claims:**

I have checked all of the proofs in the appendix. There a few points I don't understand. As I said before, I don't understand what $\sigma\_{DP}^2$ is in Theorem 3.
When I check the proof I see that it appears in the proof of the intermediate Lemma 3, in line 635. If I understand correctly you have
$$\begin{align}
    \theta\_{t+1}
    &=P\_X(\theta\_t-\gamma\_{t+1}clip\_c(\nabla\ell(\theta\_t;Z\_{t+1})))\text{ by line 129} \\
    &=P\_X(\theta\_t-\gamma\_{t+1}\tilde{\nabla}g(\theta\_t))\text{ by line 591}
\end{align}$$
So no need for $\sigma_{DP}^2$? Do I miss something?

---

> ### Author Rebuttal · Authors · 2025-03-31
>
> We thank you for your comments and careful review. Our point-by-point replies are listed below.
>
> > The definition of $\sigma_{DP}^2$ in Theorem 3?
>
> We apologize for the careless typo. Indeed, Theorem 3 should be presented without $\sigma_{DP}^2$. This term is actually introduced later in (14) and is included in the analysis of L579-597 that generalizes Theorem 3.
>
> > Additional bounded gradient assumption vs [Perdomo et al., 2020].
>
> We made sure that the assumptions needed are justified for common performative prediction problem setups. For the bounded gradient assumption (A3), we have only required $\nabla \ell$ to be bounded over $\theta \in {\cal X}$.
>
> - For strongly-convex $\ell$: as we focus on **projected** clipped SGD, if ${\cal X}$ is compact, then A3 is naturally implied using the  Lipschitz gradient assumption (A2). As such, we respectfully disagree with "assuming both smoothness and bounded gradient is too strong" since the first implies the latter in compact sets. Note that this projection-based algo, however, prompted us to develop a more general analysis than [Perdomo et al., 2020] & [Mendler-Dunner et al., 2021].
> - For non-convex $\ell$, loss functions such as sigmoid loss satisfy A3, and such condition is needed also in SOTA results such as [Li and Wai, 2024].
>
> > Additional Assumptions on bounded gradients (A3) and boundedness of $\ell$ (A7) used compared with [Koloskova et al., 2023, Theorem 3.3].
>
> We emphasize that the inclusion of distribution shift $\beta>0$ has led to significant challenge in our proof for the non-convex $\ell$ case. It is unfair to directly compare our result with that in [Koloskova et al., 2023] which assumes $\beta=0$.
>
> To see the challenge, consider a special case with $c \gg 1$ where there is no clipping. The PCSGD with ${\cal X} = \mathbb{R}^d$ is then reduced to SGD-GD in [Mendler-Dunner et al., 2021]. As discussed in [Li and Wai, 2024] and references therein, analyzing SGD-GD **without strongly convex $\ell$** has not been successfully done without extra assumptions such as assuming nonlinear least square loss, or bounded $\ell$ as in [Li and Wai, 2024]. In this regard, we only use a similar set of assumptions as [Li and Wai, 2024]. We also remark that bounded gradient assumption is also used in [Zhang et al., 2020b] for clipped SGD analysis.
>
> Lastly, as explained above, the set of assumptions for the non-convex case are justified too, e.g., with sigmoid loss as $\ell$.
>
>
>
> > Experiment: Compare PCSGD and DiceSGD with more algorithms? Conduct more real-world experiments where training data involve human input that responds strategically to the model.
>
> We conducted an additional experiment comparing PCSGD, DiceSGD with DP-Clip21-GD algorithm proposed in [Khirirat et al., 2023], see [https://ibb.co/DD9FxR71  ] -- we adopted $n=1$ & set the other param.s according to (65) in [Khirirat et al., 2023]. Observe that DiceSGD achieves better convergence.
>
> We fully agree with the necessity to conduct real world experiments. However, as an academic paper focusing on the theoretical understanding of performative behavior, we are unable to provide real world experiments at the moment due to limited resources. We notice that [Perdomo et al., 2020; Hardt & Mendler, 2023] also reuse standard static datasets and design distribution shift dynamics like we do, e.g., in the logistics regression example on the GiveMeSomeCredits dataset.
>
> There is indeed a challenge in designing real world experiments as such experiments would inevitably involve human subjects, where prior approval for compliances with ethical standards will be needed. Overcoming such challenge is beyond the aim of this research whose goal is to advance theories of performative prediction. Nevertheless, it is a promising research direction for the community.

---

### Decision · Program_Chairs · 2025-05-01

**Decision:**

Accept (poster)

**Comment:**

Summary: This paper studies the convergence of clipped SGD algorithms with decision-dependent data distribution. In particular, the paper characterizes the stochastic bias with projected clipped SGD (PCSGD) algorithm, which is caused by the clipping operator that prevents PCSGD from reaching a stable solution (focusing on both strongly convex and non-convex problems). In addition, the paper proposes remedies to mitigate the bias, either by utilizing an optimal step size design for PCSGD. Numerical experiments verifying the theoretical results are also presented.

On Reviews: All reviewers gave a score 3 (weak accept) for this work. This classifies the paper as borderline. However, by reading the reviews and going through the paper, i believe the strengths outweigh the weaknesses. The weaknesses/questions mentioned in the reviewers were handled nicely by the authors during the rebuttals as well.

I find the paper well-written, with clear and easily understandable contributions. Studying the convergence of clipped SGD algorithms with decision-dependent data distribution is a timely topic. I advise the authors to incorporate the feedback they received into the updated version of their work. I recommend acceptance.